# Chronic intermittent hypoxia reveals role of the Postinspiratory Complex in the mediation of normal swallow production

**Alyssa D Huff[1], Marlusa Karlen-Amarante[1], Luiz M Oliveira[1], Jan-Marino Ramirez[1,2]***

[1]Center for Integrative Brain Research, Seattle Children's Research Institute, Seattle, United States; [2]Department of Neurological Surgery, University of Washington School of Medicine, Seattle, United States

*For correspondence:
jan.ramirez@seattlechildrens.org

**Competing interest:** The authors declare that no competing interests exist.

**Abstract** Obstructive sleep apnea (OSA) is a prevalent sleep-related breathing disorder that results in multiple bouts of intermittent hypoxia. OSA has many neurological and systemic comorbidities, including dysphagia, or disordered swallow, and discoordination with breathing. However, the mechanism in which chronic intermittent hypoxia (CIH) causes dysphagia is unknown. Recently, we showed the postinspiratory complex (PiCo) acts as an interface between the swallow pattern generator (SPG) and the inspiratory rhythm generator, the preBötzinger complex, to regulate proper swallow-breathing coordination (Huff et al., 2023). PiCo is characterized by interneurons co-expressing transporters for glutamate (Vglut2) and acetylcholine (ChAT). Here we show that optogenetic stimulation of ChATcre:Ai32, Vglut2cre:Ai32, and ChATcre:Vglut2FlpO:ChR2 mice exposed to CIH does not alter swallow-breathing coordination, but unexpectedly disrupts swallow behavior via triggering variable swallow motor patterns. This suggests that glutamatergic–cholinergic neurons in PiCo are not only critical for the regulation of swallow-breathing coordination, but also play an important role in the modulation of swallow motor patterning. Our study also suggests that swallow disruption, as seen in OSA, involves central nervous mechanisms interfering with swallow motor patterning and laryngeal activation. These findings are crucial for understanding the mechanisms underlying dysphagia, both in OSA and other breathing and neurological disorders.

## eLife assessment

This **important** study represents a follow-up of previous papers by Huff et al. (2023) in which the authors further investigate a specific medullary region named the postinspiratory complex (PiCo) involved in the control of swallow behavior and its coordination with breathing. In the present work, they tested the impact of chronic intermittent hypoxia on the swallow motor pattern evoked by optogenetic stimulation of the same medullary area in transgenic mice. These **solid** results indicate that in chronic intermittent hypoxia-exposed mice PiCo stimulation triggers atypical swallow motor patterns. The experimental procedures are rigorous and technically remarkable. The work will be of interest in the field of respiratory physiology and pathophysiology since a disruption of swallowing and possibly discoordination with breathing may be involved in diseases characterized by the presence of hypoxic conditions such as obstructive sleep apnea.

## Introduction

Obstructive sleep apnea (OSA) is highly prevalent and a major public health concern (*Chang et al., 2023*; *McNicholas et al., 2015*; *Phillipson, 1993*; *Ramirez et al., 2013*; *Roberts et al., 2022*; *Wright*

*and Sheldon, 1998*). It is characterized by frequent bouts of apnea during sleep caused by pharyngeal collapse, resulting in multiple bouts of hypoxia referred to as chronic intermittent hypoxia (CIH). CIH increases the gain of the carotid body response to hypoxia, which seems to be a major cause for the multiple comorbidities of OSA (*Iturriaga, 2023*; *Prabhakar et al., 2023*). These OSA-related comorbidities (*Pack, 2023*) include an increase in mortality (*Vgontzas et al., 2023*) and cancer risk (*Sánchez-de-la-Torre et al., 2023*), increased arousal and sleep fragmentation (*Horner, 2023*), increased sympathetic drive leading to cardiovascular disease, metabolic syndromes such as obesity and diabetes (*Kurnool et al., 2023*), renal disease, asthma (*Bonsignore et al., 2019*), and decreased cognition (*Brockmann and Gozal, 2022*). OSA is also commonly associated with altered and disordered swallow function, clinically known as dysphagia (*Pizzorni et al., 2021*; *Schindler et al., 2014*). Clinical studies have begun to investigate physiological parameters of OSA-related dysphagia (*Bhutada et al., 2022*; *de Luccas and Berretin-Felix, 2021*), but little is known about the underlying mechanisms.

CIH and the increased gain in carotid body activity lead to disturbances in multiple neuronal mechanisms originating in the central nervous system (*Arias-Cavieres et al., 2021*; *Arias-Cavieres et al., 2020*; *da Silva et al., 2021*; *Domingos-Souza et al., 2021*; *Jia et al., 2022*; *Kline, 2010*; *Kline et al., 2019*; *Lin et al., 2007*; *Marciante et al., 2021*; *Ramirez et al., 2020*; *Souza et al., 2019*). CIH directly affects neuronal network functions within the ventral respiratory column (VRC), in particular the preBötzinger complex (preBötC) (*Garcia et al., 2017*; *Garcia et al., 2016*), a critical rhythmogenic network implicated in swallow-breathing coordination (*Huff et al., 2022*).

Swallows share anatomical structures with breathing, and it is critical these two behaviors are coordinated to prevent aspiration of food/liquid into the airway. Dysphagia, or disruption of swallow and discoordination with breathing, is directly linked to altered quality of life and failure to thrive in respiratory-related diseases such as OSA (*Bhutada et al., 2020*; *Kato et al., 2016*; *Pizzorni et al., 2021*; *Schindler et al., 2014*) and chronic obstructive pulmonary disease (COPD) (*Garand et al., 2018*; *Ghannouchi et al., 2016*; *Nagami et al., 2017*), and neurodegenerative diseases such as Parkinson's disease (*Troche et al., 2010*), Alzheimer's disease (*Priefer and Robbins, 1997*), motor neuron diseases (*Walshe, 2014*), and aging (*Ashley et al., 2006*).

Swallow-breathing coordination depends on the precise temporal activation of the pharyngeal and laryngeal muscles, as well as muscles involved in respiratory control. This coordination is controlled by various regions throughout the brainstem. The generation of swallow is thought to be governed by the caudal portion of the nucleus tractus solitaries (cNTS), specifically the interstitial and intermediate portions (*Altschuler et al., 1989*; *Kessler and Jean, 1985a*; *Kessler and Jean, 1985b*), presumably the swallow pattern generator (SPG). In rodents, swallow predominately occurs during a respiratory phase referred to as postinspiration, the transitory phase from inspiration to expiration (*Huff et al., 2023*). Thus, activity in the cNTS must be coordinated with the inspiratory rhythm generator, the preBötC (*Smith et al., 1991*), and the postinspiratory rhythm generator, the postinspiratory complex (PiCo) (*Anderson et al., 2016*), to prevent swallows from occurring during inspiration increasing the risk for aspiration.

Recently published studies have demonstrated that PiCo acts as an interface for swallow and laryngeal postinspiratory behaviors for proper coordination and timing of swallow and breathing (*Huff et al., 2023*). In the present study, we explored the effects of CIH on PiCo and its role in coordinating swallowing and breathing in order to understand how OSA and other disorders associated with intermittent hypoxia (e.g., epilepsy, Rett syndrome) lead to dysphagia. Continuing on from experiments preformed in control mice exposed to room air (*Huff et al., 2023*), we studied the impact of CIH using an established mouse model for OSA (*Garcia et al., 2017*; *Garcia et al., 2016*; *Huff et al., 2022*; *Peng and Prabhakar, 2004*; *Peng et al., 2021*). Similar to our control model, optogenetic stimulation of PiCo in ChATcre:Ai32, Vglut2cre:Ai32, and ChATcre:Vglut2FlpO:ChR2 mice stimulated both swallow and laryngeal activation. Unexpectedly, we find that PiCo-triggered swallow-breathing coordination itself is not altered, rather the alteration is in the swallow motor pattern. We propose that PiCo is involved in swallow motor patterning and CIH disrupts connections between PiCo and the SPG.

## Results

### Optogenetic stimulation of neurons in the PiCo region

Previously we demonstrated that optogenetic stimulation of PiCo neurons triggers swallow and laryngeal activation when exposed to room air (*Huff et al., 2023*). However, when exposed to CIH,

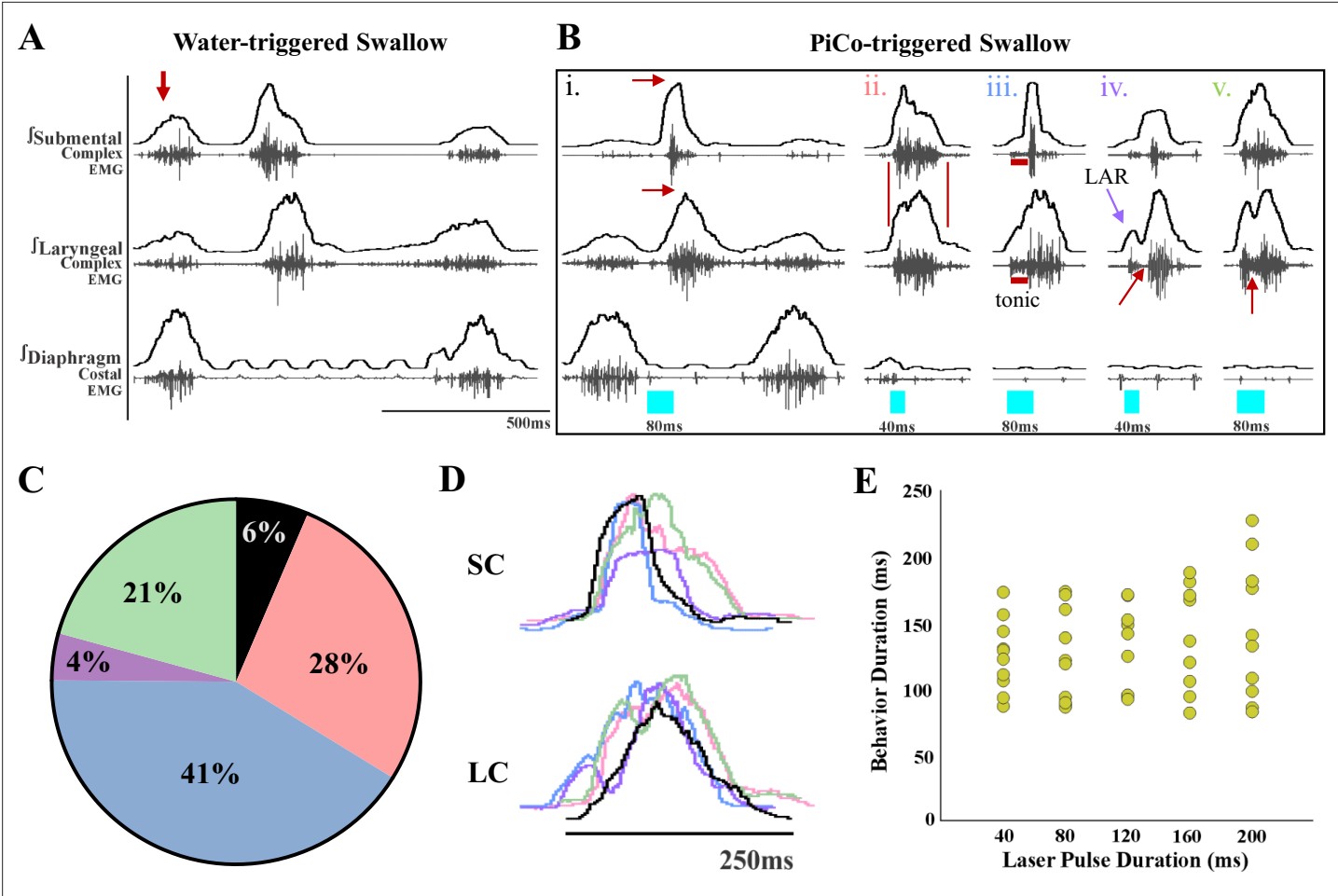

**Figure 1.** Optogenetic stimulation of postinspiratory complex (PiCo)-specific ChATcre:Vglut2FlpO:ChR2 neurons triggers variable swallow motor patterns in mice exposed to chronic intermittent hypoxia (CIH). (**A**) Representative trace of water-triggered swallow. (**B**) Representative traces of PiCo-triggered swallows: (**Bi**) 'Classic' swallow with the preserved rostrocaudal sequence shown in the red arrows. (**Bii**) 'Non-classic' swallow with similar onset, offset, and loss of sequence in submental and laryngeal complexes. (**Biii**) 'Tonic' pre-swallow activity with preserved rostrocaudal sequence and low tonic submental and laryngeal activity during the laser pulse, converging into a swallow. (**Biv**) 'Laryngeal adductor reflex' (LAR) (blue arrow) followed by a swallow. There is a period of quiescent activity between the LAR and swallow (red arrow). (**Bv**) 'Non-LAR' followed by a swallow. There is an absence of quiescent activity between the laryngeal activity and the swallow (red arrow). (**C**) Percentage of all PiCo-triggered swallows (816 total swallows) in ChATcre:Vglut2FlpO:ChR2 mice. Black is classic, pink is non-classic, blue is tonic, purple is LAR, and green is non-LAR. (**D**) Representative traces of submental complex (SC) and laryngeal complex (LC) from the swallows in (**B**) with color coding the same as (**C**). (**E**) Scatter plot of behavior duration versus laser pulse duration for swallow in ChATcre:Vglut2FlpO:ChR2 mice (N = 11). Each gold dot represents the average duration per mouse.

The online version of this article includes the following figure supplement(s) for figure 1:

**Figure supplement 1.** No significant differences in swallow-breathing characteristics between water-triggered swallows and postinspiratory complex (PiCo)-triggered swallows in mice exposed to chronic intermittent hypoxia (CIH).

**Figure supplement 2.** Chronic intermittent experimental protocol.

optogenetic stimulation of ChATcre:Vglut2FlpO:ChR2 neurons triggered a variety of abnormal swallow motor patterns (*Figure 1B*). Only 6% of all PiCo-triggered swallows could be characterized as normal, classic swallows (*Figure 1Bi*), while the vast majority of swallow motor patterns had atypical shapes and temporal sequences. This is potentially problematic since precise temporal muscle activation during swallowing is necessary for pharyngeal clearance to ensure a patent airway (*Pitts, 2014*). We characterized these atypical swallow patterns as follows: (1) non-classic swallow, 28%: submental and laryngeal shape, onset, and offset are similar (*Figure 1Bii*). (2) Tonic pre-swallow, 41%: low-amplitude tonic activity of the submental and laryngeal complexes during the laser pulse with a swallow immediately following (*Figure 1Biii*). (3) Laryngeal adductor reflex (LAR) + swallow 4%: quick burst of submental and laryngeal complex followed by a short quiescence in activity then a swallow

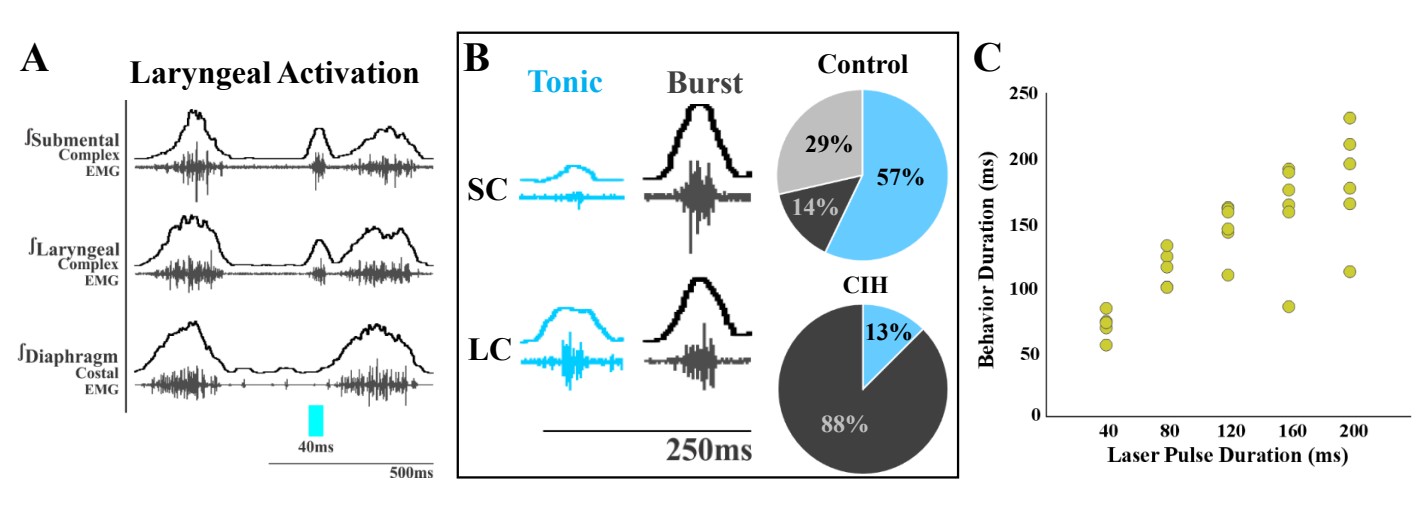

**Figure 2.** Optogenetic stimulation of postinspiratory complex (PiCo)-specific ChATcre:Vglut2FlpO:ChR2 neurons stimulates submental complex burst during laryngeal activation in mice exposed to chronic intermittent hypoxia (CIH). (**A**) Representative traces of PiCo-stimulated laryngeal activation with burst pattern submental complex activity. (**B**) Representative traces of laryngeal activation-related submental complex activity patterns, tonic and burst, and percent of each mouse with the corresponding pattern in control and CIH mice. In control ChATcre:Vglut2FlpO:ChR2 mice, four mice had tonic submental complex activity, one burst activity, and two no submental activity (*Huff et al., 2023*). In CIH-exposed, one mouse had tonic activity and seven burst submental activity. (**C**) Scatter plot of behavior duration versus laser pulse duration for laryngeal in ChATcre:Vglut2FlpO:ChR2 CIH-exposed mice. Each gold dot represents the average duration per mouse.

(*Figure 1Biv*). (4) Non-LAR + swallow, 21%: quick burst of submental and laryngeal complex followed by low-amplitude activity merging into a swallow (*Figure 1Bv*). A mixed-effects ANOVA detected a significant effect on swallow duration due to stimulation duration (p-value=0.03); however, Tukey's multiple-comparisons test revealed no significant differences in swallow behavior duration across stimulation durations (*Figure 1E*). Regardless of the motor pattern, swallow duration is independent of laser pulse duration, each are considered a swallow and will be grouped as swallows for further analysis.

We also observed differences in laryngeal activation when exposed to CIH. Optogenetic stimulation of ChATcre:Vglut2FlpO:ChR2 neurons did not stimulate laryngeal activation in 3 of the 11 mice. In the eight mice where laryngeal activation was stimulated, a burst of submental complex activity was present during PiCo-stimulated laryngeal activation in seven mice, while one mouse had a low-amplitude tonic activity (*Figure 2B*). Looking back to the control mice exposed to room air (*Huff et al., 2023*), only one out of seven mice had a burst of submental complex activity, four mice had a low tonic submental complex activity, and two mice had no submental activity. A mixed-effects ANOVA detected a significant effect on laryngeal activation duration due to stimulation duration (p-value=0.01) in CIH conditions. Tukey's multiple-comparisons test revealed significant differences ($p<0.05$) between 40 ms and 80 ms, 120 ms, 160 ms, and 200 ms, indicating laryngeal activation duration is dependent on laser pulse duration (*Figure 2C*).

## Probability of triggering a swallow

We next compared the probability of triggering a swallow between all three genetic types exposed to CIH. There were no PiCo-triggered swallows in 4 out of 14 mice in response to optogenetic stimulation of ChATcre:Ai32. In Vglut2cre:Ai32, no PiCo-triggered swallows in 3 out of 11 mice. However, stimulation of ChATcre:Vglut2FlpO:ChR2 neurons triggered a swallow in all 11 mice. The mechanism in which swallow was never triggered in some of the ChATcre:Ai32 and Vglut2cre:Ai32 CIH-exposed mice is unknown. However, it is appropriate to suggest CIH alters activity of these neuronal types since these variable responses between genetic mouselines is not seen under room air conditions (*Huff et al., 2023*).

A two-way ANOVA revealed a significant difference between temporal characteristics and the genetically defined neuron type (p<0.0001, df = 4, $F$ = 17.37) in ChATcre:Ai32 (N = 14), Vglut2cre:Ai32

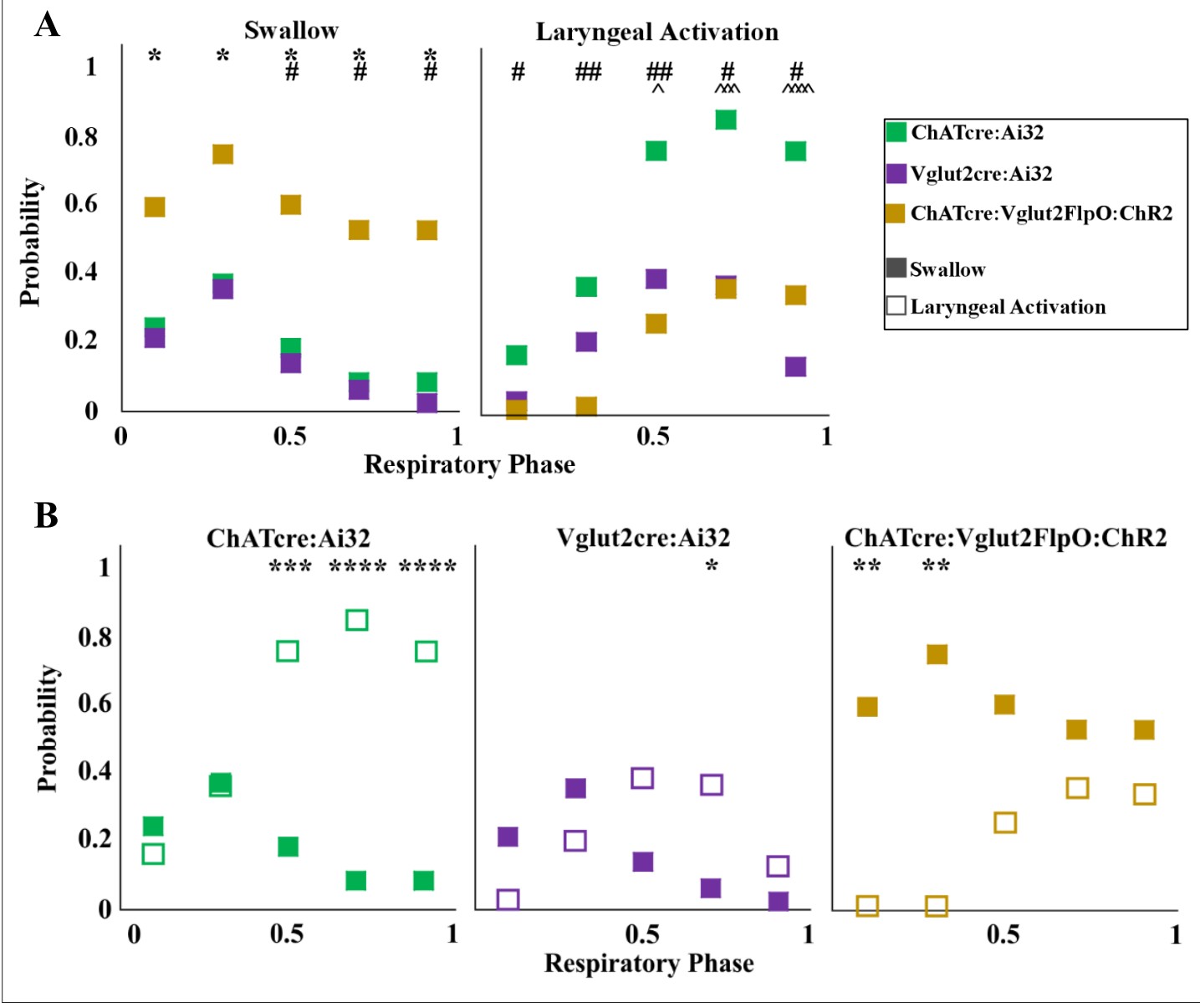

**Figure 3.** Optogenetic stimulation of postinspiratory complex (PiCo) neurons in mice exposed to chronic intermittent hypoxia (CIH) regulates swallow and laryngeal activation in a phase-specific manner. (**A**) Scatter plot of the probability of triggering a swallow (left) or laryngeal activation (right) across the respiratory phase (0 start of inspiration, 1 start of next inspiration) in ChATcre:Ai32 mice (green) Vglut2cre:Ai32 mice (purple), and ChATcre:Vglut2FlpO:ChR2 mice exposed to CIH (gold). * indicates significant difference in probability between Vglut2cre:Ai32 and ChATcre:Vglut2FlpO:ChR2, # indicates significant difference in probability between ChATcre:Ai32 and ChATcre:Vglut2FlpO:ChR2, and ^ indicates significant difference in probability between ChATcre:Ai32 and Vglut2cre:Ai32. (**B**) Scatter plot of the probability of triggering a swallow (closed square) versus laryngeal activation (open square) in all three genetic types exposed to CIH.

The online version of this article includes the following figure supplement(s) for figure 3:

**Figure supplement 1.** Postinspiratory complex (PiCo)-triggered swallows reset the respiratory rhythm, while non-swallows have minimal effect, a concept not altered by chronic intermittent hypoxia (CIH).

(N = 11), and ChATcre:Vglut2FlpO:ChR2 (N = 11) when looking at the probability of triggering a swallow (*Figure 3A*). A post hoc Tukey's multiple-comparison test revealed there is no difference in the probability of triggering a swallow between ChATcre:Ai32 and Vglut2cre:Ai32 mice. However, there in an increased probability of triggering a swallow when ChATcre:Vglut2FlpO:ChR2 neurons are activated within 50% (p=0.04), 70% (p=0.03), and 90% (p=0.02) of the respiratory cycle compared to ChATcre:Ai32. There is also an increased probability of triggering a swallow when

ChATcre:Vglut2FlpO:ChR2 neurons are activated at all phases of the respiratory cycle: 10% (p=0.03), 30% (p=0.04), 50% (p=0.02), 70% (p=0.02), and 90% (p=0.01) compared to Vglut2cre:Ai32. Whereas under control conditions, when mice were exposed to room air, we found no significant difference in the probability of triggering a swallow between all three genetic types (*Huff et al., 2023*). This indicates the importance of further evaluation on all three CIH-exposed genetic mouse types.

## Probability of triggering laryngeal activation

Optogenetic stimulation of ChATcre:Ai32 and Vglut2:Ai32 neurons stimulated laryngeal activation in all mice exposed to CIH. However, in ChATcre:Vglut2FlpO:ChR2 CIH mice, laryngeal activation was never stimulated in 3 out of 11 mice.

A two-way ANOVA revealed a significant difference between temporal characteristics and the genetically defined neuron type (p<0.0001, df = 4, *F* = 31.98) in ChATcre:Ai32, Vglut2cre:Ai32, and ChATcre:Vglut2FlpO:ChR2 CIH-exposed mice with regards to the probability of triggering laryngeal activation (*Figure 3A*). A post hoc Tukey's multiple-comparison test revealed there is no difference in the probability of triggering laryngeal activation between Vglut2cre:Ai32 and ChATcre:Vglut2FlpO:ChR2 CIH-exposed mice. However, there is an increased probability of triggering laryngeal activation when ChATcre:Ai32 neurons are activated within 50% (p=0.03), 70% (p=0.0002), and 90% (p<0.0001) of the respiratory cycle compared to Vglut2cre:Ai32 CIH-exposed mice. There is an increased probability of triggering laryngeal activation when ChATcre:Ai32 neurons are activated at all phases of the respiratory cycle: 10% (p=0.03), 30% (p=0.002), 50% (p=0.002), 70% (p=0.01), and 90% (p=0.05) compared to ChATcre:Vglut2FlpO:ChR2 CIH-exposed mice.

## PiCo phase-dependent response

Stimulation of PiCo region triggers a swallow or stimulates laryngeal activation in a respiratory phase-dependent manner (*Huff et al., 2023*; *Figure 3*). However, this stimulation-evoked phase dependency differs among the three CIH-exposed genetically defined neuron types. This difference is likely due to varying effects CIH has on each genetic mouse, with swallows never being triggered in four ChATcre:Ai32 and three Vglut2cre:Ai32, as discussed above. A two-way ANOVA revealed a significant interaction between time and behavior (*p*<0.0001, df = 4, *F* = 10.99) in ChATcre:Ai32, Vglut2cre:Ai32, and ChATcre:Vglut2:ChR2 mice (*Figure 3B*). A post hoc Tukey's multiple-comparison test revealed laryngeal activation is stimulated with a significantly higher probability when CIH-exposed ChATcre:Ai32 neurons are activated at 50% (p=0.0005), 70% (p<0.0001), and 90% (p<0.0001) of the respiratory cycle. Laryngeal activation is stimulated with a significantly higher probability when CIH-exposed Vglut2cre:Ai32 neurons are activated at 70% (p=0.03) of the respiratory cycle. However, swallow is triggered with a significantly higher probability when CIH-exposed ChATcre:Vglut2FlpO:ChR2 neurons are activated within the first 10% (p=0.004) and 30% (p=0.001) of the respiratory cycle.

## Respiratory response to optogenetic stimulation of PiCo

We divided PiCo stimulated responses into either swallow or non-swallow (*Figure 3—figure supplement 1*). Stimulation of PiCo neurons that resulted in either laryngeal activation or in a 'no-motor response' were considered non-swallows. Using a Pearson correlation and simple linear regression, the correlation coefficient (r, *Figure 3—figure supplement 1A*) and line of best fit (slope, *Figure 3—figure supplement 1B*), respectively, was calculated for each CIH-exposed genetic mouse type and response to determine the degree of correlation between behavior response and reset of the respiratory rhythm. This test reveals that there is a high degree of correlation between shifting or delaying the following inspiratory burst and triggering a swallow when stimulating ChATcre:Ai32 (*r* = 0.76, p<0.0001, slope = 0.75), Vglut2cre:Ai32 (*r* = 0.71, p<0.0001, slope = 0.82), and ChATcre:Vglut2FlpO:ChR2 (*r* = 0.79, p<0.0001, slope = 0.79) CIH-exposed mice. This suggests that triggering a swallow in each genetic type has a strong effect on resetting the respiratory rhythm in CIH conditions. These results add to the current understanding that swallow has a hierarchical control over the respiratory rhythm.

We found a moderate degree of correlation between the following inspiratory burst and non-swallows stimulated in ChATcre:Ai32 (*r* = 0.36, p<0.0001, slope = 0.16) and a low degree of correlation in Vglut2cre:Ai32 (*r* = 0.22, p<0.0001, slope = 0.17) and ChATcre:Vglut2FlpO:ChR2 (*r* = 0.28,

p=0.0001, slope = 0.18) mice. This suggests that triggering a swallow has a stronger effect on resetting the respiratory rhythm than activating non-swallows in all the genetic mouse types exposed to CIH.

## Swallow-related characteristics in water-triggered and PiCo-triggered swallows

*Figure 1A* depicts the swallow motor patterns of a water-evoked swallow and *Figure 1B* of various swallow motor patterns of PiCo-evoked swallow. A mixed-effect ANOVA revealed no significant difference in swallow onset relative to inspiratory onset (*Figure 1—figure supplement 1A*) and swallow onset relative to inspiratory peak (*Figure 1—figure supplement 1B*) between PiCo-evoked and water-evoked swallows in CIH-exposed mice. We were unable to perform a repeated-measures ANOVA due to swallows not being triggered by PiCo stimulation in some mice, as mentioned above. All water- and PiCo-triggered swallow-related characteristics in all three CIH-exposed genetic mouse lines are reported in *Supplementary file 1*.

PiCo-triggered swallows are characterized by a significant decrease in duration compared to swallows evoked by water in ChATcre:Ai32 (265 ± 132 ms vs 144 ± 101 ms; paired *t*-test: p=0.0001, *t* = 5.21, df = 8), Vglut2cre:Ai32 (308 ± 184 ms vs 125 ± 44 ms; paired *t*-test: p=0.0003, *t* = 6.46, df = 7), and ChATcre:Vglut2FlpO:ChR2 (230 ± 67 ms vs 130 ± 35 ms; paired *t*-test: p=0.0005, *t* = 5.62, df = 8) mice exposed to CIH (*Supplementary file 1*).

PiCo-triggered swallows have a significant decrease in submental amplitude compared to swallows evoked by water in ChATcre:Ai32 (91 ± 7 vs 38 ± 35% of max; paired *t*-test: p=0.002, *t* = 4.91, df = 7), Vglut2cre:Ai32 (84 ± 10 vs 45 ± 32% of max; paired *t*-test: p=0.006, *t* = 3.84, df = 7), and ChATcre:Vglut2FlpO:ChR2 (88 ± 10 vs 39 ± 22% of max; paired *t*-test: p=0.001, *t* = 7.47, df = 8) CIH-exposed mice (*Supplementary file 1*).

### Sex-specific differences in swallows triggered by optogenetic stimulation of PiCo region

All water- and PiCo-triggered sex-specific swallow-related characteristics in all three CIH-exposed genetic mouse lines are reported in *Supplementary files 2–4*. In ChATcre:Ai32 female mice, PiCo-triggered swallow onset relative to inspiratory onset occurs later in the respiratory cycle (0.31 ± 0.04 vs 0.37 ± 0.04; paired *t*-test: p=0.04, *t* = 2.38, df = 8). There are no sex-specific differences in PiCo-triggered swallows in Vglut2cre:Ai32 or ChATcre:Vglut2FlpO:ChR2 CIH-exposed mice.

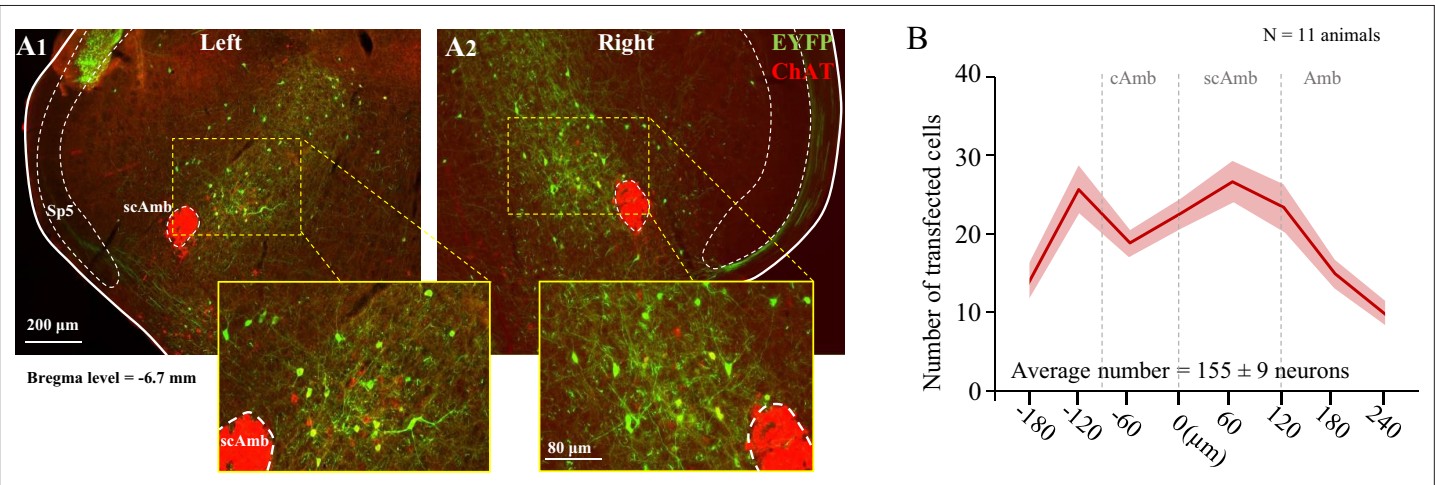

**Figure 4.** Transfection of cholinergic/glutamatergic neurons in postinspiratory complex (PiCo) in ChATcre:Vglut2FlpO:ChR2 chronic intermittent hypoxia (CIH)-exposed mice. (**A**) Transverse hemisection through Bregma level (–6.7 mm) of the transfected neurons into PiCo bilaterally, left (**A1**) and right (**A2**), with the pAAV-hSyn Con/Fon hChR2(H134R)-EYFP vector. (**B**) Rostrocaudal distribution of the total number of transfected neurons counted 1:2 series of 25 μm sections into PiCo of 11 animals with an average of 155 ± 9 SEM neurons. Amb, nucleus ambiguus; cAmb, nucleus ambiguus pars compacta; scAmb, nucleus ambiguus pars semi-compacta.

## Neuroanatomy of PiCo transfection

Post hoc histological analysis was performed in the double-conditioned ChATcre:Vglut2FlpO:ChR2 mouse to check the transfection of PiCo neurons after injection of the pAAV-hSyn Con/Fon hChR2(H134R)-EYFP vector (ChATcre:Vglut2FlpO:ChR2) (*Figure 4*). NAmb cholinergic neurons had no transfection and the rostrocaudal distribution of the transgene-expressing neurons was analyzed. Eleven ChATcre:Vglut2FlpO:ChR2 mice were stimulated and triggered a swallow, while laryngeal activation was triggered in only eight mice. In all 11 CIH-exposed mice, we found that 155 ± 9 neurons expressed EYFP (*Figure 4*). There was a high bilateral transfection of ChR2 in PiCo neurons ranging from 121 to 189 neurons with one animal having 88 neurons. The animal with 88 neurons did not appear to have a different optogenetic response compared to the other animals. CIH does not appear to have an effect on viral transfection of ChATcre:Vglut2FlpO neurons since room air-exposed mice had an average of 133 ± 16 neurons that expressed EYFP (*Huff et al., 2023*).

## Discussion

In mice, and other mammals including humans, exposed to room air, swallow-related muscles follow a stereotypic, rostro-caudal sequential muscle pattern activation (*Basmajian and Dutta, 1961*; *Doty and Bosma, 1956*; *Ertekin and Aydogdu, 2003*; *Pitts and Iceman, 2023*; *Thexton et al., 2007*). The activation pattern must be precisely timed and shaped to guarantee that food and liquid particles are guided into the esophagus and ultimately the digestive system, rather than the lungs. Disturbances in any aspect of the swallow patterning could lead to aspiration, resulting in the penetration of food or liquid through the vocal folds, entering the lungs. In the long term, aspiration may lead to aspiration pneumonia. We found that after exposure to CIH only 6% of swallows triggered by optogenetic stimulation of PiCo, specifically the ChATcre:Vglut2FlpO:ChR2 neurons, followed this classic swallow motor pattern (*Figure 1Bi*), while 28% of PiCo-triggered swallows lost important characteristics of the sequential motor activation crucial for proper bolus transport (*Pitts and Iceman, 2023*). In these 'non-classic swallows', the submental and laryngeal muscles were activated simultaneously, which could impair the effective transport of food/liquid from the oral cavity into the esophagus and digestive system. Stimulation of glutamatergic/cholinergic neurons also induced 'tonic pre-swallow' activity of both the submental and laryngeal complexes prior to triggering a swallow in 41% of PiCo-triggered swallows. While fine-wire EMG studies are an excellent evaluation tool to observe force and temporal motor pattern of sequential swallow-related muscles, it must be combined with tools such as video-fluoroscopic swallow study (VFSS) and/or high-resolution manometry (HRM) in order to characterize the functional significance of these alterations to the swallow motor pattern shown in this study (*Park et al., 2017*). Since the preparation in this study utilizes only fine-wire EMGs, we are not able to evaluate or comment on the functional significance of the variable swallow motor patterns. However, it is appropriate to suggest the variable alterations in the swallow motor pattern seen in CIH-exposed mice will alter the effectiveness of the swallow to clear the pharynx and maintain a patent airway.

While we do not intend to make direct quantitative comparisons between the previously published PiCo-triggered swallows in control mice exposed to room air (*Huff et al., 2023*) and the data presented here for mice exposed to CIH, we believe it is important to compare the conclusions made in these two studies. There is a higher probability of triggering a swallow when PiCo is activated during inspiration or immediately after during postinspiration. Whereas laryngeal activation is more probable when PiCo is activated further into expiration. This remains true for both ChATcre:Vglut2FlpO:ChR2 control mice exposed to room air (*Huff et al., 2023*) and mice exposed to CIH (*Figure 3*). However, there is a decreased probability of PiCo-triggered swallows in ChATcre:Ai32 and Vglut2:Ai32 exposed to CIH, unlike the mice exposed to room air. The mechanism in which CIH affects the ChATcre:Ai32 and Vglut2cre:Ai32 neurons is unknown and would benefit from further exploration into their neuronal properties following CIH exposure. We observed a high degree of correlation between shifting or delaying the following inspiratory burst and triggering a swallow in both control mice (*Huff et al., 2023*) and mice exposed to CIH (*Figure 3—figure supplement 1*) indicating swallow's hierarchical control over respiratory rhythm generators. We also observed no differences between the water-triggered swallow onset and PiCo-triggered swallow onset in relation to inspiration onset and peak inspiration (*Figure 1—figure supplement 1*). Thus, CIH does not alter PiCo's ability to coordinate

swallow and breathing. Rather, our data reveal CIH disrupts the swallow motor sequence, which is likely due to changes in the interaction between PiCo and the SPG, presumably located in the cNTS.

While it has previously been demonstrated that PiCo is an important region in swallow-breathing coordination (*Huff et al., 2023*), previous studies did not demonstrate that PiCo is involved in swallow motor patterning itself. Here we show for the first time that CIH leads to disturbances in the generation of the swallow motor pattern that is activated by stimulating PiCo. This suggests that PiCo is not only important for coordinating swallow and breathing, but also modulating swallow motor patterning. Further studies are necessary to directly evaluate the presumed interactions between PiCo and the cNTS.

## Variability to swallow motor pattern

Previously we suggested that the PiCo-evoked laryngeal activation could be a central and integral component to the LAR (*Huff et al., 2023*). However, there had been no reports of centrally evoked LAR or an LAR independent of mechanical, electrical, or chemical peripheral stimulation or swallow. Here we show, for the first time, optogenetic stimulation of PiCo, in mice exposed to CIH, triggers swallow-related LAR (*Figure 1Biv*). The afferent limb of the LAR is governed by the internal branch of the superior laryngeal nerve (SLN) where sensory impulses travel through the nodose ganglion and terminate on the cNTS (*Ambalavanar et al., 2004*; *Sessle, 1973*). It is likely that both swallow and LAR involves interneurons of the solitarius-ambiguus pathway, possibly activated by PiCo stimulation (*Ambalavanar et al., 2004*; *Mifflin, 1993*).

In addition to the classic LAR response, PiCo stimulation also triggered non-LAR swallows (*Figure 1Bv*). To be considered an LAR, there must be a quiescence period of laryngeal activity between the reflex and the swallow (*Ambalavanar et al., 2004*; *Ludlow et al., 1992*). In these swallows, there was a convergence of the initial laryngeal peak and augmenting swallow-related laryngeal activity. The functional merging of these two activity patterns is not understood. It has been shown that excessive or inappropriate laryngeal activity could lead to functional disorders or life-threatening conditions (*Sun et al., 2011*), such as obstructive apnea, laryngospasm, spasmodic dysphonia, asphyxia, sudden infant death syndrome, and aspiration pneumonia (*Ikari and Sasaki, 1980*; *Ludlow et al., 1995*; *Thompson et al., 2005*; *Wang et al., 2016*). While we are unable to evaluate the functional significance of the variable swallow motor patterns triggered by PiCo in this study, we previously reported CIH causes a disruption in swallow motor pattern with a delay in swallow-related laryngeal activation. This same delay was also generated when preBötC Dbx1 neurons were stimulated (*Huff et al., 2022*).

The known rostrocaudal swallow motor sequence can be modulated due to changes in sensory feedback (*King et al., 2020*). However, here we are stimulating a central microcircuit, believed to not activate sensory components of the SPG (*Huff et al., 2023*), which induces great variability to this motor sequence when exposed to CIH. Our study indicates that PiCo neurons are highly integrated with the overall swallow motor pattern. Variability in response to CIH on swallow motor pattern is reminiscent to the increased variability also seen in the generation of the respiratory motor pattern (*Garcia et al., 2017*; *Garcia et al., 2016*). To understand the complex disruption of CIH on the swallow motor pattern, it is important to note that we are measuring changes in two muscle complexes, which spread among three motor neuron pools: hypoglossal nucleus, trigeminal nucleus, and nucleus ambiguus (*Badran et al., 2005*; *Bieger and Neuhuber, 2006*; *Kemplay and Cavanagh, 1983*; *Razlan et al., 2018*). There was no statistical difference in the probability of triggering a swallow during optogenetic stimulation of ChATcre:Ai32, Vglut2cre:Ai32, and ChATcre:Vglut2FlpO:ChR2 neurons in mice exposed to room air (*Huff et al., 2023*). However, when exposed to CIH, ChATcre:Ai32 and Vglut2:Ai32 mice have a lower probability of triggering a swallow – in some mice swallow was never triggered via PiCo activation, while water-triggered swallows remained – compared to the ChATcre:Vglut2FlpO:ChR2 mice. While it is possible that portions of the presumed SPG remain less affected by CIH, which could offset these instabilities to produce functional swallows, our data suggest that PiCo targets microcircuits within the SPG that are highly affected by CIH. The NTS is a primary first site for upper airway and swallow-related sensory termination in the brainstem (*Jean, 1984*). CIH induces changes to the cardio-respiratory Vglut2 neurons, resulting in an increase in cNTS neuronal activity (*Kline, 2010*; *Kline et al., 2007*), as well as changes to preBötC neurons (*Garcia et al., 2017*; *Garcia et al., 2016*) and ChAT neurons in the basal forebrain (*Tang et al., 2020*). It is reasonable to suggests that CIH has differential effects on neurons that

only express ChATcre and Vglut2cre versus the PiCo-specific interneurons that co-express ChATcre and Vglut2FlpO, emphasizing the importance of targeting and manipulating these PiCo-specific interneurons.

## Variability to laryngeal activation motor pattern

Acute bouts of extreme hypoxia reduce the excitability of laryngeal adductor neurons, suggesting a 'fail-safe' mechanism that in the presence of hypoxia LAR is prevented (*Ikari and Sasaki, 1980*). However, acute bouts of intermittent hypoxia elicit long-term facilitation of the recurrent laryngeal nerve (RLN), but do not recruit additional postinspiratory laryngeal muscle activity or augmented LAR (*Bautista et al., 2012*). After exposure of CIH, we observed a shift in the dominance of 'laryngeal activation' submental complex motor activity pattern. Under control conditions, optogenetic stimulation of ChATcre:Vglut2FlpO:ChR2 neurons resulted in laryngeal activation with three types of submental activation: tonic, burst, and no activation, with the majority of mice having tonic submental activation (*Figure 2B*). However, when exposed to CIH, seven out of eight mice had burst submental activation. This is consistent with the known increased activity of pharyngeal dilator muscles in OSA patients (*Mezzanotte et al., 1992*; *Saboisky et al., 2012*) and in CIH animal models (*Kubin, 2019*). This could constitute as a compensatory mechanism to keep the airway patent (*Dempsey et al., 2010*).

## Limitations

The use of anesthesia continues to be an unavoidable limitation for this preparation. There were two mice that did not swallow in response to water stimulation, most likely due to anesthetic depth. In this preparation, we are unable to directly determine the functionality of the variable swallow motor patterns seen after CIH. Different experimental techniques such as videofluoroscopy would need to be used to directly evaluate functional significance. This technique is beyond the scope of this study and not possible to perform in this preparation. We acknowledge this limits our ability to make direct comparisons between dysphagic swallows in OSA patients. In addition, this preparation does not allow for recording of PiCo neurons to evaluate the direct effects of CIH in PiCo neuronal activity. These limitations are a trade-off for the unique advantages of manipulating specific genetically defined neuron types under in vivo conditions to assess neuronal alterations in OSA-related dysphagia.

## Conclusion

Clinical dysphagia is typically seen in OSA, and other disorders associated with CIH. OSA-related dysphagia has been characterized as a delayed swallow reflex, decreased time between swallow and initiation of the next inspiration, and disruption of the pharyngeal phase of swallow (*Levring Jäghagen et al., 2003*; *Teramoto et al., 1999*; *Valbuza et al., 2011*). Using an in vivo mouse model, we recently described that CIH delays swallow-related laryngeal activation that increases the risk for foreign materials to enter into the airway (aspiration) instead of the esophagus (*Huff et al., 2022*). Here we show that PiCo, a neuronal network that is critical for the generation of postinspiratory activity (*Anderson et al., 2016*) and implicated in the coordination of swallowing and breathing (*Huff et al., 2023*), is severely affected by CIH. Stimulating PiCo-specific interneurons after CIH exposure evoked swallows characterized by abnormal swallow motor patterns, while the swallow-breathing coordination was relatively unaffected. The severe disruption in the precise temporal sequences of swallows evoked by PiCo stimulation after CIH exposure has important implications for understanding the mechanisms underlying dysphagia. Our data suggest that PiCo is not only a neuronal structure critical for regulating postinspiratory and swallow activity, but also critical for the patterning of swallow itself. To the best of our knowledge, no microcircuit has previously been identified that can directly impact the swallow motor pattern itself. Swallows are characterized by a very robust, yet complex stereotypic motor sequence that is typically triggered in an all-or-none manner. The strict temporal sequence of this motor pattern is critical for allowing the physiological transport of food to reach the digestive system. Any disturbance in this sequence has detrimental consequences including aspiration, the leading cause of death in many neurodegenerative diseases. Our study also suggests that OSA-related dysphagia may not primarily be due to swallow-breathing discoordination, but rather involve a central nervous dysfunction of the swallow pattern and laryngeal activation.

# Materials and methods

## Animals

Adult (P51-148, average P76) male and female mice were bred at Seattle Children's Research Institute (SCRI) and used for all experiments. Vglut2-IRES-cre and ChAT-IRES-cre homozygous breeder lines were obtained from Jackson Laboratories (stock numbers 028863 and 031661, respectively). Cre mice were crossed with homozygous mice containing a floxed STOP channelrhodopsin-2 fused to an EYFP (Ai32) reporter sequence from Jackson Laboratories (stock number 024109). Vglut2-IRES-cre crossed with Ai32 will be reported as Vglut2:Ai32 and the ChAT-IRES-cre crossed with Ai32 as ChAT:Ai32. ChAT-IRES-cre, and Vglut2-IRES2-FlpO-D, approved gene name 129S-Slc17a6$^{tm1.1(flpo)}$$_{Hze}$/J, were obtained from Jackson Laboratories (#031661 and #030212, respectively). To generate double-transgenic mice, the ChATcre and Vglut2FlpO strains were interbred to generate compound homozygotes expressing both ChATcre and Vglut2FlpO and will be reported as ChATcre:Vglut2FlpO. Mice were randomly selected from the resulting litters by the investigators for the control protocol (*Huff et al., 2023*) or the CIH protocol (present study). Offspring were group housed with ad libitum access to food and water in a temperature-controlled (22 ± 1°C) facility with a 12 hr light/dark cycle. All experiments and animal procedures were approved by the SCRI's Animal Care and Use Committee (protocol 00058) and were conducted in accordance with the National Institutes of Health and ARRIVE guidelines (*Percie du Sert et al., 2020*).

## Brainstem injection of AAV

For the AAV injections, we target the PiCo neurons, previously described (*Anderson et al., 2016*; *Huff et al., 2023*), and confirmed by the present results (*Figure 4*). We restricted ChR2 expression to the PiCo region in order to transfect and photo-stimulate the region with the highest density of ChATcre:Vglut2FlpO neurons in PiCo region (*Anderson et al., 2016*). For AVV injection, the mice were anesthetized with isoflurane (2%). The correct plane of anesthesia was assessed by the absence of the corneal and hind-paw withdrawal reflexes. Mice received postoperative ketoprofen (7 mg/kg, subcutaneous [s.c.]) for two consecutive days. All surgical procedures were performed under aseptic conditions. The hair over the skull and neck were removed and skin disinfected. The mice were then placed prone on a stereotaxic apparatus (bite bar set at –3.5 mm for flat skull; David Kopf Instruments Tujunga, CA). A 0.5-mm-diameter hole was drilled into the occipital plate on both sides caudal to the parieto-occipital suture. Viral solutions were loaded into a 1.2 mm internal diameter glass pipette broken to a 20 µm tip (external diameter). To target the PiCo region with ChR2-AAV, the pipette was inserted in the brainstem in the following coordinates: 4.8 mm below the dorsal surface of the cerebellum, 1.1 mm lateral to the midline, and 1.6 mm caudal to the lambda and bilateral injections of 150 nl were made slowly at 50 nl/min, using a glass micropipette and an automatic nanoliter injector (NanoinjectII, Drummond Scientific Co., Broomall, PA). The mouse was allowed to recover for 3 d before beginning the CIH protocol (see below).

The mouse strain containing ires-cre and ires-FlpO in ChAT$^+$ and Vglut2$^+$, respectively, had successful transfection of PiCo neurons by using a pAAV-hSyn Con/Fon hChR2(H134R)-EYFP adenovirus vector (Cat# 55645-AAV8; AddGene, USA; abbreviated as AAV8-ConFon-ChR2-EYFP), herein named ChATcre:Vglut2FlpO:ChR2 in this study. This AAV is a cre-on/FlpO-on ChR2-EYFP under the synapsin promoter and encoded the photoactivatable cation channel channelrhodopsin-2 (ChR2, H134R) fused to EYFP. The vector was diluted to a final titer of $1 \times 10^{13}$ viral particles/ml with sterile phosphate-buffered saline.

## Chronic intermittent hypoxia

Mice of the ChATcre:Ai32, Vglut2cre:Ai32, and ChATcre:Vglut2FlpO:ChR2 were kept in collective cages with food and water ad libitum placed inside custom-built chambers (volume: 185 l) equipped with gas injectors as well as oxygen ($O_2$) sensors (Oxycycler, Huff Technologies Inc). This study was done in parallel with the previous published control study where one chamber was used for CIH (current study) and the other for control (*Huff et al., 2023*). The CIH group was exposed to intermittent episodes of hypoxia, continuous injection of nitrogen ($N_2$) for 60 s, in order to reduce the percentage of inspired $O_2$ inside the chamber from 21% to 4.5–5%. Then continuous injection of compressed air for 5 min into the chamber to return the percentage of $O_2$ to 21% before the start of a new hypoxia cycle. Compressed air and $N_2$ injection into the chambers were regulated by a valve system, automatically

operated by customized software (Oxycycler, Huff Technologies Inc). This protocol was repeated with 80 bouts per day (8 hr) during the light cycle in a 12 hr light/dark cycle room, for an average of 21 d. Of note the range was 10–29 d, but internal analysis of pilot 10-day protocol showed no difference from the 21-day protocol and were combined in this study. In the remaining 16 hr, the mice were kept under normoxia condition (21% $O_2$). Control mice were kept in a replicated chamber under normoxic conditions (21% $O_2$), 24 hr a day during the same duration as the CIH protocol (*Figure 1—figure supplement 2*). The mice under control conditions have been published (*Huff et al., 2023*).

## In vivo experiments

The same experimental protocol was performed for all Vglut2cre:Ai32, ChATcre:Ai32, and ChATcre:Vglut2FlpO:ChR2 mice. Adult mice were initially anesthetized with 100% $O_2$ and 1.5% isoflurane (Aspen Veterinary Resources Ltd, Liberty, MO) for 2–3 min in an induction chamber. Once the breathing slowed, they were injected with urethane (1.5 g/kg, i.p. Sigma-Aldrich, St. Louis, MO) and secured supine on a custom surgical table. Core temperature was maintained through a water heating system (PolyScience, Niles, IL) built into the surgical table. Mice were then allowed to spontaneously breathe 100% $O_2$ for the remainder of the surgery and experimental protocol. Adequate depth of anesthesia was determined via heart and breathing rate, as well as lack of toe pinch response every 15 min. A supplemental dose of 0.01 ml of urethane was given to maintain adequate anesthetic depth, when necessary. Bipolar electromyogram (EMG) electrodes were placed in the costal diaphragm to monitor respiratory rate and heart rate throughout the experiment. The trachea was exposed through a midline incision and cannulated caudal to the larynx with a curved (180°) tracheal tube (PTFE 24G, Component Supply, Sparta, TN). The hypoglossal (XII) and vagus (X) nerves were then dissected followed by cannulation of the trachea. The RLN was carefully dissected away from each side of the trachea before the cannula was tied in and sealed with super glue to ensure no damage to the RLN. The trachea and esophagus were then cut to detach the rostral end of the trachea just caudal to the cricoid cartilage, preserving the arytenoids and bilateral RLN. A tube filled with 100% $O_2$ was attached to the cannulated trachea to provide supplemental oxygen throughout the experiment. Continuing in the supine position, the occipital bone was removed, followed by continuous perfusion of the ventral medullary surface with warmed (~36°C) artificial cerebral spinal fluid (aCSF; in mM: 118 NaCl, 3 KCl, 25 NaHCO$_3$, 1 NaH$_2$PO$_4$, 1 MgCl$_2$, 1.5 CaCl$_2$, 30 D-glucose) equilibrated with carbogen (95% $O_2$, 5% $CO_2$) by a peristaltic pump (Dynamax RP-1, Rainin Instrument Co, Emeryville, CA). As previously published (Figure 6a in *Huff et al., 2022*), the XII and X nerves were isolated unilaterally, cut distally, and their activity was recorded from a fire-polished pulled borosilicate glass tip (B150-86-15, Sutter Instrument, Novato, CA) filled with aCSF connected to the monopolar suction electrode (A-M Systems, Sequim, WA) and held in a 3D micromanipulator (Narishige, Tokyo, Japan). Multiple bipolar EMGs, using 0.002″ and 0.003″ coated stainless steel wires (A-M Systems, part nos. 790600 and 791000, respectively), simultaneously recorded activity from several swallow and respiratory-related muscle sites. According to the techniques of *Basmajian and Stecko, 1962*, the electrodes were placed using hypodermic needles 30G (part no. 305106, BD Precision Glide, Franklin Lakes, NJ) in the *submental complex*, which consists of the geniohyoid, mylohyoid, and anterior digastric muscles, to determine swallow activity. The *laryngeal complex*, consisting of the posterior cricoarytenoid, lateral, transverse, and oblique arytenoid, cricothyroid and thyroarytenoid muscles, to determine laryngeal activity during swallow, as well as postinspiratory activity (*Figure 1—figure supplement 2C*). The *costal diaphragm* is used to measure the multifunctional activity for both inspiration, as well as *Schluckatmung*, a less common diaphragmatic activation during swallow activity (*Pitts et al., 2018*). Glass fiber optic (200 um diameter) connected to a blue (447 nm) laser and DPSS driver (Opto Engine LLC, Salt Lake City, UT) was placed bilaterally in light contact with the ventral surface of the brainstem overtop of the predetermined PiCo (*Anderson et al., 2016*; *Figure 1—figure supplement 2*). At the end of the experiment, mice were euthanized by an overdose of anesthetic followed by rapid decapitation or trans-cardial perfusion (see 'Histology').

## Stimulation protocols

First, swallow was stimulated by injecting 0.1 cc of water into the mouth using a 1.0 cc syringe connected to a polyethylene tube. Second, 25 pulses of each 40 ms, 80 ms, 120 ms, 160 ms, and 200 ms continuous TTL laser stimulation at PiCo were repeated, at random, throughout the respiratory

cycle. The lasers were each set to 0.75 mW and triggered using Spike2 software (Cambridge Electronic Design, Cambridge, UK). These stimulation protocols were performed in all ChATcre:Ai32, Vglut2cre:Ai32, and ChATcre:Vglut2FlpO:ChR2.

## Analysis

All electroneurogram (ENG) and EMG activity were amplified and band-pass filtered (0.03–1 kHz) by a differential AC Amplifier (A-M System model 1700), acquired in an A/D converter (CED 1401; Cambridge Electronic Design) and stored using Spike2 software (Cambridge Electronic Design). Using the Spike2 software, data was further processed using a band-pass filtered (200–700 Hz, 40 Hz transition gap), then rectified and smoothed (20 ms). Using the Spike2 software, the ECGdelete 02.s2s script was used to remove heart artifact, when present, from the ENG and EMG recordings.

We evaluated swallows that were trigged by injection of water into the mouth as well as behaviors in response to laser stimulation applied to the PiCo region: swallow, laryngeal activation, and no motor response. Swallow was characterized as a delayed response to the laser outlasting and independent of the laser duration, activation of XII, X, submental, and laryngeal complex. Diaphragm activity during PiCo-triggered swallows (*schluckatmung*) was present in some animals but not all. Laryngeal activation was characterized as activity of the XII, X, and laryngeal complex dependent on laser pulse duration, and absence of the diaphragm EMG activity. The submental complex was active in a tonic or burst pattern during laryngeal activation. No response was characterized as lack of motor response to the laser and was grouped with laryngeal activation for the non-swallow analysis in respiratory phase shift plots (*Figure 3—figure supplement 1*). Previously published swallow-related parameters were used to look at swallow-breathing characteristics (Figure 6 in *Huff et al., 2022*) *Swallow duration* was determined by the onset to the termination of the submental complex EMG activity. In case the submental complex muscles were not available, then it was determined by the onset to the termination of the XII ENG activity. *Swallow sequence* was calculated as the time difference between the peak activation of the laryngeal and submental complex EMG activity. *Schluckatmung* duration was determined by the onset to the termination of the diaphragm EMG activity during a swallow. *Laryngeal activation duration* was determined by the onset to the termination of the laryngeal complex EMG activity. *Diaphragm inter-burst interval* was calculated as the offset of the diaphragm EMG activity to the onset of the subsequent breath. *Inspiratory delay* was calculated as the offset of the swallow-related laryngeal complex EMG activity to the onset of the subsequent breath. Duration and amplitude of each nerve and muscle were determined by the onset to the termination of that respective nerve/muscle activity during swallow. All durations are reported in milliseconds, and all amplitudes are reported as a '% of max' calculated as the % of the maximum baseline (water swallow) amplitude.

As previously reported (Figure 6d in *Huff et al., 2022*), respiratory phase reset curves calculated by defining the respiratory cycle as the onset of the diaphragm to the onset of the subsequent diaphragm activity. The *phase shift* elicited by each stimulation of water was calculated as the duration of the respiratory cycle containing the stimulus, divided by the preceding respiratory cycle. The phase of the swallow stimulation *(respiratory phase)* was calculated as the time between the onset of the inspiration (diaphragm) and the stimulus onset, divided by the expected phase. The average phase shift was then plotted against the respiratory phase in bins containing 1/10 of the expected phase (*Baertsch et al., 2018*). Line graphs of swallow frequency in relation to inspiration were created by the phase of breathing in which swallow occurred in, calculated as the onset of inspiration to the onset of swallow divided by the respiratory cycle duration and plotted against the number of swallows that occurred within the 1/10 binned respiratory phase (*swallow onset: insp onset)*. Swallow was also plotted in relation to the peak activation of the diaphragm as a duration with zero equaling the peak of the inspiratory related diaphragm activity (*swallow onset: insp peak)*.

Probability plots were calculated by assigning a '0' to the no response behavior or a '0 or 1' to the laryngeal activation or swallow behavior. These numbers were then averaged and plotted against the *respiratory phase* and binned to 1/10 of the respiratory phase.

All data are expressed as mean ± standard deviation (SD), unless otherwise noted. Statistical analyses were performed using GraphPad Prism 9 (GraphPad Software, Inc, La Jolla). Differences were considered significant at $p < 0.05$. Investigators were not blinded during analysis. Sample sizes were chosen on the basis of previous studies.

## Histology

At the end of experiments, animals were deeply anesthetized with 5% isoflurane in 100% oxygen and perfused through the ascending aorta with 20 ml of phosphate-buffered saline (PB; pH 7.4) followed by 4% phosphate-buffered (0.1 M; pH 7.4; 20 ml) paraformaldehyde (Electron Microscopy Sciences, Fort Washington, PA). The brains were removed and stored in the perfusion fixative for 4 hr at 4°C, followed by 20% sucrose for 8 hr. A series of coronal sections (25 µm) from the brains were cut using a cryostat and stored in cryoprotectant solution at –20°C (20% glycerol plus 30% ethylene glycol in 50 ml phosphate buffer, pH 7.4) prior to histological processing. All histochemical procedures were done using free-floating sections.

Choline acetyltransferase (ChAT) was detected using a polyclonal goat anti-ChAT antibody (AB144P; Millipore; 1:100) and EYFP was detected using a polyclonal mouse anti-GFP (06–896, Millipore; 1:1000) diluted in PB containing 2% normal donkey serum (017-000-121, Jackson ImmunoResearch Laboratories) and 0.3% Triton X-100, and incubated for 24 hr. Sections were subsequently rinsed in PB and incubated for 2 hr in an Alexa 594 donkey anti-goat (705-585-003; 1:250; Jackson ImmunoResearch Laboratories) and Alexa 488 donkey anti-mouse (715-545-150; 1:250; JacksonImmuno Research Laboratories). For all secondary antibodies used, control experiments confirmed that no labeling was observed when primary antibodies were omitted. The sections were mounted on slides in a rostrocaudal sequential order, dried, and covered with fluoromount (00-4958-02; Thermo Fisher). Coverslips were affixed with nail polish.

Sections were also examined to confirm the transfected cells. Section alignment between specimens was done relative to a reference section. The rostral segment of PiCo was identified by the last section with the caudal end of the facial motor neurons and the first section with the rostral portion of the inferior olives. To distinguish PiCo in each section, we used the nucleus ambiguus (Amb), the inferior olives (IO), and the ventral spinocerebellar tract (vsc) as the main anatomic structures. The section that contains the rostral portion of Amb (more densely packed, i.e., cAmb) is the section that contains the rostral portion of PiCo, in a caudal direction, the compacta portion of Amb turns into a semi-compacta portion (scAmb), being aligned as the zero point in the rostral-caudal graphs. Further caudal, the scAmb turns in the non-compacta portion of Amb (*Akins et al., 2017*; *Baertsch et al., 2018*; *Kottick et al., 2017*; *Vann et al., 2018*), characterizing the caudal edge of PiCo. PiCo was also anatomically characterized by immunohistological labeling, revealing ChAT-positive neurons located dorsomedial to c-scAmb and caudal to the facial nucleus as previously described (*Toor et al., 2019*; *Anderson et al., 2016*). As shown in *Figure 4*, according to the *Paxinos and Franklin, 2019* mouse atlas, the transfected cells were located slightly dorsal to the NAmb near Bregma level –6.84 mm, ~1100 µm from the midline, and ~700 µm above the marginal layer.

## Cell counting, imaging, and data analysis

A VS120-S6-W Virtual Slide Scanner (Olympus) was used to scan all the sections. Images were taken with a color camera (Nikon DS-Fi3). To restrict any influences on our counted results, the photomicrography and counting were performed by one blind researcher. ImageJ (version 1.41; National Institutes of Health, Bethesda, MD) was used for cell counting and Canvas software (ACD Systems, Victoria, Canada, v. 9.0) was used for line drawings. A one-in-two series of 25 µm brain sections was used per mouse, which means that each section analyzed was 50 µm apart. The area analyzed was delimited based on previous reports (*Anderson et al., 2016*) (mean of 5423 µm²). The sections were counted bilaterally, averaged, and the numbers reported as mean ± standard error of the mean (SEM). Section alignment were relative to a reference section, as previously described (*Anderson et al., 2016*) and based on *Paxinos and Franklin, 2019*.

## Acknowledgements

We are grateful to receive the NIH grants P01 HL14454 and Project 2 (awarded to JMR) HL144801 (awarded to JMR), R01 HL151389 (awarded to JMR), and F32 HL160102-01 (awarded to AH) for funding this project.

## Additional information

### Funding

| Funder | Grant reference number | Author |
|---|---|---|
| National Institutes of Health | HL144801 | Jan-Marino Ramirez |
| National Institutes of Health | HL151389 | Jan-Marino Ramirez |
| National Institutes of Health | HL160102 | Alyssa D Huff |
| National Institutes of Health | P01 HL14454 | Jan-Marino Ramirez |
| National Institutes of Health | RO1 HL126523 | Jan-Marino Ramirez |

The funders had no role in study design, data collection and interpretation, or the decision to submit the work for publication.

### Author contributions

Alyssa D Huff, Luiz M Oliveira, Conceptualization, Data curation, Formal analysis, Methodology, Writing - original draft, Writing – review and editing; Marlusa Karlen-Amarante, Conceptualization, Data curation, Methodology, Writing – review and editing; Jan-Marino Ramirez, Conceptualization, Funding acquisition, Methodology, Writing - original draft, Writing – review and editing

### Author ORCIDs

Alyssa D Huff (iD) http://orcid.org/0000-0003-2817-251X
Marlusa Karlen-Amarante (iD) http://orcid.org/0000-0002-4733-3035
Jan-Marino Ramirez (iD) http://orcid.org/0000-0002-5626-3999

### Ethics

All experiments and animal procedures were approved by the Seattle Children's Research Institute's Animal Care and Use Committee (Protocol 00058) and were conducted in accordance with the National Institutes of Health and ARRIVE guidelines (Percie du Sert et al., 2020).

Reviewer #1 (Public Review): https://doi.org/10.7554/eLife.92175.3.sa1
Reviewer #2 (Public Review): https://doi.org/10.7554/eLife.92175.3.sa2
Author response https://doi.org/10.7554/eLife.92175.3.sa3

# Additional files

### Supplementary files

• Supplementary file 1. Means, standard deviations (SD), p-values, $t$-statistic ($t$), degrees of freedom (df), from a paired $t$-test; and the direction of change for swallow-related parameters when evoked by water (water swallows) and optogenetic stimulation of PiCo in (**A**) ChATcre:Ai32, (**B**) Vglut2cre:Ai32, and (**C**) ChATcre:Vglut2FlpO:ChR2 mice.

• Supplementary file 2. Means, standard deviations (SD), p-values, $F$-value, $t$-statistic ($t$), degrees of freedom (df), from an unpaired $t$-test; and the direction of change for swallow-related parameters between male and female mice during water swallows and PiCo-stimulated swallows in ChATcre:Ai32 mice.

• Supplementary file 3. Means, standard deviations (SD), p-values, $F$-value, $t$-statistic ($t$), degrees of freedom (df), from an unpaired $t$-test; and the direction of change for swallow-related parameters between male and female mice during water swallows and PiCo-stimulated swallows in Vglut2cre:Ai32 mice.

• Supplementary file 4. Means, standard deviations (SD), p-values, $F$-value, $t$-statistic ($t$), degrees of freedom (df), from an unpaired $t$-test; and the direction of change for swallow-related parameters between male and female mice during water swallows and PiCo-stimulated swallows in

ChATcre:Vglut2FlpO:ChR2 mice.
• MDAR checklist

### Data availability
All data is publicly available at https://doi.org/10.6084/m9.figshare.24777798.

The following dataset was generated:

| Author(s) | Year | Dataset title | Dataset URL | Database and Identifier |
|---|---|---|---|---|
| Huff A, Karlen-Amarante M, Oliveira LM, Ramirez JM | 2024 | Huff et al 2024 Experimental Data set | https://doi.org/10.6084/m9.figshare.24777798 | figshare, 10.6084/m9.figshare.24777798 |

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
