## [Editor Report · eLife assessment]

This **important** study represents a follow-up of previous papers by Huff et al. (2023) in which the authors further investigate a specific medullary region named the postinspiratory complex (PiCo) involved in the control of swallow behavior and its coordination with breathing. In the present work, they tested the impact of chronic intermittent hypoxia on the swallow motor pattern evoked by optogenetic stimulation of the same medullary area in transgenic mice. These **solid** results indicate that in chronic intermittent hypoxia-exposed mice PiCo stimulation triggers atypical swallow motor patterns. The experimental procedures are rigorous and technically remarkable. The work will be of interest in the field of respiratory physiology and pathophysiology since a disruption of swallowing and possibly discoordination with breathing may be involved in diseases characterized by the presence of hypoxic conditions such as obstructive sleep apnea.

---

## [Referee Report · Reviewer #1 (Public Review)]

Summary:

Authors were attempting to determine the extent that CIH altered swallowing motor function; specifically, the timing and probability of the activation of the larygneal and submental motor pools. The paper describes a variety of different motor patterns elicited by optogenetic activation of individual neuronal phenotypes within PiCo in a group of mice exposed to CIH. They show that there are a variety of motor patterns that emerge in CIH mice; this is apparently different than the more consistent motor patterns elicited by PiCo activation in normoxic mice (previously published)

Strengths:

The preparation is technically challenging and gives valuable information related to the role of PiCo in the pattern of motor activation involved in swallowing and its timing with phrenic activity. Genetic manipulations allow for the independent activation of the individual neuronal phenotypes of PiCo (glutamatergic, cholinergic) which is a strength.

Weaknesses:

(1) Comparisons made between experimental data acquired currently with those previously published are extremely problematic, with the potential confounding influence of changing environments, genetics and litter effects. For example, were the current mice tested at the same time as those exposed to normoxia? Are they littermates (or at least from the same colony) as those previously examined? If they were tested at the same time and age, then the authors should explicitly state this in the methods. The authors have provided no statistical analyses to determine whether there is an effect of CIH on the motor patterns. In short, how can they be sure that the phenomena they observe with respect to motor patterns is due to CIH?

(2) The data are descriptive in nature, reporting only differences (diversity) of motor patterns in this cohort of animals exposed to CIH. There is limited mechanistic insight into how PiCo manipulation alters the pattern and probability of motor activation. Can they utilize Fos or marker of activation within the nTS or other regions to provide initial insight? Or in another nucleus that contributes as part of the circuit.

(3) The differences between the genotypes (ChaTcre; Vglut2Cre; ChatCre:Vglut2FlpO) with regard to the probability of generating a swallow are not sufficiently discussed, in my view. If, as the authors state, it is "reasonable to suggest that CIH differentially affects" these populations, then what are some viable reasons? What are the known differences in these populations of neurons that could lead to variable responses? Do they project to different places?

(4) The Results section is difficult to follow and interpret. It would be beneficial to have a couple of sentences after each sub-section stating what the data actually mean. As of now it reads like a statistical report of the data with little "basic" interpretation of the data.

(5) I have a hard time understanding the functional significance of calculating and plotting the degree of correlation between shifting/delaying the following inspiratory burst and triggering a swallow.

---

## [Referee Report · Reviewer #2 (Public Review)]

The manuscript has been revised according to Reviewer's suggestions. Recommendations for the Authors have been almost entirely followed. However, there are some points where the authors state that they have made changes, but the text does not show this. The revised version would have gained in clarity if it was with track changes and numbered rows. In particular, I cannot see the following changes:

Lines 104-105: Did you mean: "We confirmed that optogenetic stimulation of PiCo neurons in ChATcre:Vglut2FlpO:ChR2 mice exposed to CIH triggers swallow and laryngeal activation similar to the control mice exposed to room air (Huff et al., 2023)." Otherwise, the sentence is not clear.

Thank you, this has been changed

Lines 228-232: "PiCo-triggered swallows are characterized by a significant decrease in duration compared to swallows evoked by water in ChATcre:Ai32 mice (265 {plus minus} 132ms vs 144 {plus minus} 101ms; paired t-test: p = 0.0001, t = 5.21, df = 8), Vglut2cre:Ai32 mice (308 {plus minus} 184ms vs 125 {plus minus} 44ms; paired t-test: p = 0.0003, t = 6.46, df = 7), and ChATcre:Vglut2FlpO:ChR2 mice (230 {plus minus} 67ms vs 130 {plus minus} 35ms; paired t-test: p = 0.0005, t = 5.62, df = 8) exposed to CIH (Table S1).".

Thank you, this has been changed

Lines 283-290: "Thus, CIH does not alter PiCo's ability to coordinate the timing for swallowing and breathing. Rather, our data reveals that CIH disrupts the swallow motor sequence likely due to changes in the interaction between PiCo and the SPG, presumably the cNTS.

While it has previously been demonstrated that PiCo is an important region in swallow-breathing coordination (Huff et al., 2023), previous studies did not demonstrate that PiCo is involved in swallow pattern generation itself. Thus, here we show for the first time that CIH resulted in the instability of the swallow motor pattern activated by stimulating PiCo, suggesting PiCo plays a role in its modulation.".

Thank you, this has been changed

Line 437: Mice of the ChATcre:Ai32, Vglut2cre:Ai32 and ChATcre:Vglut2FlpO:ChR2 lines were kept in collective cages with food and water ad libitum placed inside custom-built chambers.

Thank you, this has been changed.

Overall, the manuscript has been improved.

---

## [Author Response]

The following is the authors’ response to the original reviews.

**Public Reviews:**

**Reviewer #1 (Public Review):**
Summary:The authors were attempting to determine the extent that CIH altered swallowing motor function; specifically, the timing and probability of the activation of the larygneal and submental motor pools. The paper describes a variety of different motor patterns elicited by optogenetic activation of individual neuronal phenotypes within PiCo in a group of mice exposed to CIH. They show that there are a variety of motor patterns that emerge in CIH mice; this is apparently different than the more consistent motor patterns elicited by PiCo activation in normoxic mice (previously published).Strengths:The preparation is technically challenging and gives valuable information related to the role of PiCo in the pattern of motor activation involved in swallowing and its timing with phrenic activity. Genetic manipulations allow for the independent activation of the individual neuronal phenotypes of PiCo (glutamatergic, cholinergic) which is a strength.

We thank the reviewers for acknowledging and summarizing the strengths of this study.

Weaknesses:(1) The data presented are largely descriptive in terms of the effect of PiCo activation on the probability of swallowing and the pattern of motor activation changes following CIH. Comparisons made between experimental data acquired currently and those obtained in a previous cohort of animals (possibly years before) are extremely problematic, with the potential confounding influence of changing environments, genetics, and litter effects. The statistical analyses (i.e. comparing CIH with normoxic) appear insufficiently robust. Exactly how the data were compared is not described.

Yes, we agree the data are descriptive in terms of characterizing the effect of CIH on PiCo activation. However, we would like to emphasize that the data are also mechanistic because they characterize the effects of specifically, optogenetically manipulating PiCo neurons after being exposed to CIH.

Thank you for this comment and for pointing out our misleading description in the paper. This manuscript is meant to independently characterize the effects of CIH to the response of PiCo stimulation. We are not making direct comparisons between the previously published manuscript where mice were exposed to room air. There has been no statistical analysis made between previously published control and current CIH data, since we are not making a direct comparison, only an observational comparison.

To make this clearer, and to address the reviewers concern, we have removed the room air data from figures 1E, 2C and 3A. However, we believe it is important to keep the data from mice exposed to room air in Figure 2B since we did not include this information in the previously published manuscript. It is important to point out that all mice exposed to CIH have some form of submental activity during laryngeal activation in response to PiCo stimulation. This is not the case when mice are exposed to room air only. In this figure, only descriptive analysis are presented. We adjusted our wording throughout the text, particularly in the discussion, to eliminate any confusion that we are making direct comparisons between the two studies. The following sentence has been added to the discussion “While we do not intend to make direct quantitative comparisons between the previously published PiCo-triggered swallows in control mice exposed to room air (Huff et al 2023) and the data presented here for mice exposed to CIH, we believe it is important to compare the conclusions made in these two studies.” This was the motivation for using the eLife Advance format. Since the present study demonstrates that PiCo affects swallow patterning which was not observed in the control data.

(2) There is limited mechanistic insight into how PiCo manipulation alters the pattern and probability of motor activation. For example, does CIH alter PiCo directly, or some other component of the circuit (NTS)? Techniques that silence or activation projections to/from PiCo should be interrogated. This is required to further delineate and define the swallowing circuit, which remains enigmatic.

We agree with the reviewer that our study raises many more questions than we are able to answer at the moment. This however applies to most scientific studies. Even though swallowing has been studied for many decades, the underlying circuitry remains largely enigmatic. We will continue to investigate the role of PiCo and its interaction with the NTS, in healthy and diseased states. These investigations require many different techniques, and approaches, some of which are still in development. For example, we are currently conducting experiments that silence portions of the NTS related to swallow and PiCo: ChAT/Vglut2 neurons using novel unpublished viral approaches. However, these are separate and ongoing studies beyond the scope of the current one.

To address the reviewer’s comment, we have added to the following to the limitation section: “In addition, this preparation does not allow for recording of PiCo neurons to evaluate the direct effects of CIH in PiCo neuronal activity”. The following has also been added to the discussion: “Rather, our data reveal CIH disrupts the swallow motor sequence which is likely due to changes in the interaction between PiCo and the SPG, presumably located in the cNTS. While it has previously been demonstrated that PiCo is an important region in swallow-breathing coordination (Huff et al., 2023), previous studies did not demonstrate that PiCo is involved in swallow motor patterning itself. Here we show for the first time that CIH leads to disturbances in the generation of the swallow motor pattern that is activated by stimulating PiCo. This suggests that PiCo is not only important for coordinating swallow and breathing, but also modulating swallow motor patterning. Further studies are necessary to directly evaluate the presumed interactions between PiCo and the cNTS.”

(3) The functional significance of the altered (non-classic) patterns is unclear.

Like in our original study, the preparation used to stimulate PiCo does not allow to simultaneously characterize the functional significance of swallowing. Therefore, we have included this as a limitation in the limitation section: “In this preparation we are unable to directly determine the functionality of the variable swallow motor pattern seen after CIH. Different experimental techniques, such as videofluoroscopy would need to be used to directly evaluate functional significance. This technique is beyond the scope of this study and not possible to perform in this preparation. We acknowledge this limits our ability to make direct comparisons between dysphagic swallows in OSA patients.”

**Reviewer #1 (Recommendations For The Authors):**
(1) A more rigorous experimental approach is required. Littermates should be separated and exposed to either room air or CIH at the same (or close to the same) time.

As stated above, we did not directly compare mice exposed to room air with mice exposed to CIH. Hence, we believe this is not necessary, and it would have meant repeating all the experiments already published in the original eLife paper.

(2) Robust statistical analyses are required to determine whether the effects of CIH on the pattern/probability of motor activation are required.

Since control and CIH group were not compared in this study, statistical hypothesis testing is not appropriate or applicable.

(3) Use a combination of retrograde, Cre- AAVs and Cre-dependent approaches to interrogate the circuitry to/from PiCO that forms the swallowing network. This is what is needed to push this area forward, in my view.

Thank you for this suggestion, we will consider this suggestion as we plan for future experiments. Indeed, we are in the process of developing novel approaches. However, in this context we would like to emphasize that further network investigations are exponentially more complicated given that we need to use a Flpo/Cre approach to specifically characterize the glutamatergic-cholinergic PiCo neurons. Most other laboratories that have studied PiCo have avoided this experimental complication and used only a “cre-dependent” approach. This approach is much simpler, but the data are much less specific and the conclusions sometimes misleading. Stimulating for example cholinergic neurons in the PiCo area will also activate Nucleus ambiguus neurons, stimulating glutamatergic neurons will also activate glutamatergic neurons that are not necessarily the glutamatergic/cholinergic neurons that we use to define PiCo specifically. Readers that are unfamiliar with these different approaches often miss this important difference. Hence, compared to stimulating other areas, stimulating the cholinergic-glutamatergic neurons in PiCo is much more specific than e.g. stimulating preBötzinger complex neurons. There are no markers that will specifically stimulate only preBötzinger complex neurons or neurons in the parafacial Nucleus. Unfortunately, this difference is often overlooked.

(4) It should be made more clear how each of the "non-classic" swallowing patterns could cause dysfunction - especially to the reader who is not completely familiar with the neural control of swallowing.

We agree that it would be helpful to understand the functional implications of these alterations in swallow-related motor activation, however since our approach does not allow us to use any tools to measure or evaluate functional activity it would be inappropriate to make suggestions of this type without any data to back up our conclusion. This is why we have not speculated on the functional implications. We have added the following to the discussion section of this manuscript. “While fine wire EMG studies are an excellent evaluation tool to observe temporal motor pattern of sequential swallow related muscles; it must be combined with tools such as videofluoroscopic swallow study (VFSS) and/or high resolution manometry (HRM) in order to characterize the functional significance of these alterations to the swallow motor pattern shown in this study (Park et al., 2017). Since the preparation in this study utilizes only fine wire EMGs we are not able to evaluate or comment on the functional significance of the variable swallow motor patterns. ”

Minor:The Results should be written in a way that better conveys the neurophysiological effects of the manipulations. As it stands, it reads like a statistical report on how activation of each neuronal phenotype is statistically different from each other. As such it is difficult to read and understand the salient findings.

Thank you for this insight. We have adjusted the language in the results section.

**Reviewer #2 (Public Review):**
Summary:In this study, the authors investigated the role of a medullary region, named Postinspiratory Complex (PiCo), in the mediation of swallow/laryngeal behaviours, their coordination with breathing, and the possible impact on the reflex exerted by chronic intermittent hypoxia (CIH). This region is characterized by the presence of glutamatergic/cholinergic interneurons. Thus, experiments have been performed in single allelic and intersectional allelic recombinase transgenic mice to specifically excite cholinergic/glutamatergic neurons using optogenetic techniques, while recording from relevant muscles involved in swallowing and laryngeal activation. The data indicate that in anaesthetized transgenic mice exposed to CIH, the optogenetic activation of PiCo neurons triggers swallow activity characterized by variable motor patterns. In addition, these animals show an increased probability of triggering a swallow when stimulation is applied during the first part of the respiratory cycle. They conclude that the PiCo region may be involved in the occurrence of swallow and other laryngeal behaviours. These data interestingly improve the ongoing discussion on neural pathways involved in swallow-breathing coordination, with specific attention to factors leading to disruption that may contribute to dysphagia under some pathological conditions.The Authors' conclusions are partially justified by their data. However, it should be acknowledged that the impact of the study is to a certain extent limited by the lack of knowledge on the source of excitatory inputs to PiCo during swallowing under physiological conditions, i.e. during water-evoked swallowing. Also the connectivity between this region and the swallowing CPG, a structure not well defined, or other brain regions involved in the reflex is not known.

We thank the reviewer for the comments and the strength of the paper. However, with regards to the “lack of knowledge”, we would like to emphasize that PiCo was first described in 2016, while e.g. the preBötzinger complex was described in 1991. Thus, it is not fair to assume the same level of anatomical and physiological understanding for PiCo as we became accustomed to for the preBötzinger complex. We are fairly confident that in 25 years from now, our knowledge of the in- and outputs of PiCo will be much less limited than it currently is.

Strengths:Major strengths of the manuscript:The methodological approach is refined and well-suited for the experimental question. The in vivo mouse preparation developed for this study takes advantage of selective optogenetic stimulation of specific cell types with the simultaneous EMG recordings from upper airway muscles involved in respiration and swallowing to assess their motor patterns. The animal model and the chronic intermittent hypoxia protocol have already been published in previous papers (Huff et al. 2022, 2023).The choice of the topic. Swallow disruption may contribute to the dysphagia under some pathological conditions, such as obstructive sleep apnea. Investigations aimed at exploring and clarifying neural structures involved in this behaviour as well as the connectivity underpinning muscle coordination are needed.This study fits in with previous works. This work is a logical extension of previous studies from this group on swallowing-breathing coordination with further advances using a mouse model for obstructive sleep apnea.

We thank the reviewers for acknowledging and summarizing the strengths of this study.

Weaknesses:Major weaknesses of the manuscript:The Authors should be more cautious in concluding that the PiCo is critical for the generation of swallowing itself. It remains to demonstrate that PiCo is necessary for swallowing and laryngeal function in a more physiological situation, i.e. swallow of a bolus of water or food. It should be interesting to investigate the effects of silencing PiCo cholinergic/glutamatergic neurons on normal swallowing. In this perspective, the title should be slightly modified to avoid "swallow pattern generation" (e.g. Chronic Intermittent Hypoxia reveals the role of the Postinspiratory Complex in the mediation of normal swallow production).

Thank you for pointing out that this manuscript suggest PiCo is necessary for swallow generation. We agree further interventions to silence specifically PiCo ChAt/Vglut2 neurons will be necessary to investigate this claim. Which we have begun to evaluate for a future study by developing a novel as yet unpublished approach. We have altered language throughout the text to limit the perception that PiCo is the swallow pattern generator. We have also changed the title to say: Chronic Intermittent Hypoxia reveals the role of the Postinspiratory Complex in the mediation of normal swallow production

The duration of swallows evoked by optogenetic stimulation of PiCo is considerably shorter in comparison with the duration of swallows evoked by a physiological stimulus (water). This makes it hard to compare the timing and the pattern of motor response in CIH-exposed mice. In Figure 1, the trace time scale should be the same for water-triggered and PiCo-triggered swallows. In addition, it is not clear if exposure to CIH alters the ongoing respiratory activity. Is the respiratory rhythm altered by hypoxia? If a disturbed or irregular pattern of breathing is already present in CIH-exposed mice, could this alteration interfere with the swallowing behaviour?

Thank you. We have changed the time scale so that all representative traces are on the same time scale.

We explained in the original paper (Huff et al 2023) that the significant decrease in PiCo-evoked swallow duration compared to water evoked is likely due to the absence of oral/upper airway feedback. We are not making comparisons of the effects of CIH on swallow motor pattern between water-evoked and PiCo-evoked. Rather, we are only characterizing the effects of CIH on the swallow motor pattern in PiCo-evoked swallows. The purpose of Figure 1A is to show that the rostocaudal submental-laryngeal sequence in water-evoked swallows is preserved in “canonical” PiCo-evoked swallow like is shown in the original study. While we did not measure the effects of CIH on breathing and the respiratory pattern in this study, it has been established, by others, that CIH causes respiratory muscle weakness, impaired motor control of the upper airway and variable respiratory rhythm and rhythm generation. However, when characterizing the timing of swallow in relation to inspiration (Figure 1 Figure Supplement 1) and the reset of the respiratory rhythm (Figure 3 figure supplement 1) and by observationally comparing these results with mice exposed to room air (Huff et al 2023) we do not observe any obvious differences in swallow-breathing coordination. However, a separate study in wild-type mice focusing on a characterization of swallowing via water after CIH would be better suited to achieve a better understanding of the physiological changes of swallowing after CIH. We would like to point out that this has shown in Huff et al 2022 that altering respiratory rate/pattern via activation of various preBötzinger Complex neurons does not change swallow behavior. Except in the case of Dbx1 PreBötC neuron activation, which was independent of CIH. Increasing or decreasing respiratory rate via activation of PreBötC Vgat and SST neurons did not change the swallow pattern rather it changed the timing of when swallows occurred. It has been reported before by others that swallow has a hierarchical control over breathing and has the ability to shut breathing down. We believe that the swallowing behavior is independent of respiratory pattern and alterations in breathing pattern does not necessarily affect the swallow motor pattern rather could affect the swallow timing.

**Reviewer #2 (Recommendations For The Authors):**
AbstractLines 37-41 "Here we show that optogenetic stimulation of ChATcre:Ai32, Vglut2cre:Ai32, and ChATcre:Vglut2FlpO:ChR2 mice exposed to CIH does not alter swallow-breathing coordination, but unexpectedly the generation of swallow motor pattern was significantly disturbed."It should be better:"Here we show that optogenetic stimulation of ChATcre:Ai32, Vglut2cre:Ai32, and ChATcre:Vglut2FlpO:ChR2 mice exposed to CIH does not alter swallow-breathing coordination, but unexpectedly triggers variable swallow motor patterns".

Thank you, this has been changed

Lines 41-43 "This suggests, glutamatergic-cholinergic neurons in PiCo are not only critical for the gating of postinspiratory and swallow activity but also play important roles in the generation of swallow motor pattern." I suggest removing any language claiming PiCo is swallow gating and change "generation" in "modulation""This suggests that glutamatergic-cholinergic neurons in PiCo are not only critical in regulating swallow-breathing coordination but also play important roles in the modulation of swallow motor pattern."

Thank you, this has been changed

Introduction:Line 88-90: Actually, in Huff et al. 2023 it is said "PiCo acts as an interface between the swallow pattern generator and the preBötzinger complex to coordinate swallow and breathing". Please, change accordingly. Please, remove Toor et al., 2019 since their conclusions are quite different.Line 100-101: Please, change the sentence according to the comments reported above.

Thank you, this has been changed

Results:Lines 104-105: Did you mean: "We confirmed that optogenetic stimulation of PiCo neurons in ChATcre:Vglut2FlpO:ChR2 mice exposed to CIH triggers swallow and laryngeal activation similar to the control mice exposed to room air (Huff et al., 2023)." Otherwise, the sentence is not clear.

Thank you, this has been changed

Lines 129-130: This finding is not surprising since similar results have been reported in Huff et al. 2023.

Thank you, we wanted to confirm that CIH did not alter this characteristic, which it did not. We believe that it is important to include this as it is a criterion for characterizing laryngeal activation.

Lines 219: The number of water swallows is considerably lower than stimulation-evoked swallows. Why?

We inject water into the mouth three times. Typically, there is one swallow in response to each water injection. Pico is stimulated 25 times at each duration. If we were to stimulate swallow with water as many times as optogenetic stimulation there would be an adaptive response to the water stimulation and the mouse would not respond. This does not seem to be the case with PiCo stimulation. Simple answer is, there are many more PiCo stimulations than water stimulation.

Lines 228-232: "PiCo-triggered swallows are characterized by a significant decrease in duration compared to swallows evoked by water in ChATcre:Ai32 mice (265 {plus minus} 132ms vs 144 {plus minus} 101ms; paired t-test: p = 0.0001, t = 5.21, df = 8), Vglut2cre:Ai32 mice (308 {plus minus} 184ms vs 125 {plus minus} 44ms; paired t-test: p = 0.0003, t = 6.46, df = 7), and ChATcre:Vglut2FlpO:ChR2 mice (230 {plus minus} 67ms vs 130 {plus minus} 35ms; paired t-test: p = 0.0005, t = 5.62, df = 8) exposed to CIH (Table S1).".

Thank you, this has been changed

Line 252 and 254: remove SEM.

Thank you, this has been changed

DiscussionLine 267: ...(Figure 1Bi), while 28% of PiCo-triggered swallows...

Thank you, this has been changed

Lines 283-290: "Thus, CIH does not alter PiCo's ability to coordinate the timing for swallowing and breathing. Rather, our data reveals that CIH disrupts the swallow motor sequence likely due to changes in the interaction between PiCo and the SPG, presumably the cNTS.While it has previously been demonstrated that PiCo is an important region in swallow-breathing coordination (Huff et al., 2023), previous studies did not demonstrate that PiCo is involved in swallow pattern generation itself. Thus, here we show for the first time that CIH resulted in the instability of the swallow motor pattern activated by stimulating PiCo, suggesting PiCo plays a role in its modulation.".

Thank you, this has been changed

Could the observed effects be due to a non-specific effect of hypoxia on neuronal excitability? In addition, it should be considered that PiCo-triggered swallows lack the behavioural setting of water-evoked swallows and do not activate the sensory component of the SPG to the same extent as the water-evoked swallows.

Yes, this is very possible. We stated in our first manuscript that the decrease in PiCo-triggered swallow duration, as compared to water-triggered swallow duration, is likely because oral sensory components are not being activated to the same extent (Huff et al. 2023). Since we do not directly measure neuronal excitability, it is not known (in this study) whether CIH causes changes in the excitability to swallow related areas. However, others have shown increased excitability and activity of Vglut2 neurons after CIH exposure (Kline et al 2007,2010), and we have shown e.g. changes in the excitability of preBötC neurons (Garcia et al. 2016, 2017).

Lines 293-300: The sentence is not clear. Is there any evidence indicating that glutamatergic neurons are differently affected by hypoxia than cholinergic neurons?

Thank you, these sentences have been changed to increase clarity. The section now reads: There was no statistical difference in the probability of triggering a swallow during optogenetic stimulation of ChATcre:Ai32, Vglut2cre:Ai32 and ChATcre:Vglut2FlpO:ChR2 neurons in mice exposed to room air (Huff et al 2023). However, when exposed to CIH, ChATcre:Ai32 and Vglut2:Ai32 mice have a lower probability of triggering a swallow -- in some mice swallow was never triggered via PiCo activation, while water-triggered swallows remained – compared to the ChATcre:Vglut2FlpO:ChR2 mice. While it is possible that portions of the presumed SPG remain less affected by CIH, which could offset these instabilities to produce functional swallows, our data suggest that PiCo targets microcircuits within the SPG that are highly affected by CIH. The NTS is a primary first site for upper airway and swallow-related sensory termination in the brainstem (Jean, 1984). CIH induces changes to the cardio-respiratory Vglut2 neurons, resulting in an increase in cNTS neuronal activity (Kline, 2010; Kline et al., 2007), as well as changes to preBötzinger neurons (Garcia et al., 2017; Garcia et al., 2016) and ChAT neurons in the basal forebrain (Tang et al., 2020). It is reasonable to suggests that CIH has differential effects on neurons that only express ChATcre and Vglut2cre versus the PiCo-specific interneurons that co-express ChATcre and Vglut2FlpO, emphasizing the importance of targeting and manipulating these PiCo-specific interneurons.”

Lines 372-374: "Here we show that PiCo, a neuronal network which is critical for the generation of postinspiratory activity (Andersen et al. 2016) and implicated in the coordination of swallowing and breathing (Huff et al., 2023), is severely affected by CIH.".

Thank you, this has been changed.

MethodsLine 398: Did you mean Slc17a6-IRES2-FlpO-D?

Thank you, this has been changed.

Line 399: were.

Thank you, this has been changed.

Line 403: ... expressing both ChAT and Vglut2 and will be reported as ChATcre:Vglut2FlpO.

Thank you, this has been changed.

Line 437: Mice of the ChATcre:Ai32, Vglut2cre:Ai32 and ChATcre:Vglut2FlpO:ChR2 lines were kept in collective cages with food and water ad libitum placed inside custom-built chambers.

Thank you, this has been changed.

Line 479: (Figure 6a in Huff et al., 2022).Line 497: What does Fig 7 refer to?

This should say Figure 1- figure supplement 2, This has been changed

Lines 501-506: "First, swallow was stimulated by injecting 0.1cc of water into the mouth using a 1.0 cc syringe connected to a polyethylene tube. Second, 25 pulses of each 40ms, 80ms, 120ms, 160ms and 200ms continuous TTL laser stimulation at PiCo was repeated, at random, throughout the respiratory cycle. The lasers were each set to 0.75mW and triggered using Spike2 software (Cambridge Electronic Design, Cambridge, UK). These stimulation protocols were performed in all ChATcre:Ai32, Vglut2cre:Ai32, and ChATcre:Vglut2FlpO:ChR2." .

Thank you, this has been changed.

Line 526 and 540: (Fig.6 in Huff et al., 2022) and (Fig.6d in Huff et al., 2022).

Thank you, this has been fixed

Line 594: Figure 5 doesn't exist. Please, change the sentence.

Thank you, this has been fixed

Line 595 and 609: The reference Kirkcaldie et al. 2012 is referred to the neocortex and doesn't seem appropriate. Please, quote the atlas of Paxinos and Franklin.

Thank you, this has been changed.

Reference:Please, correct throughout the text editing of references by removing e.g J.M. or A. or David D. and so on. Only surnames should be mentioned.

Thank you, this has been changed.

Figures:Figure 1. A and B as well as the purple arrow are lacking. In addition, optogenetic stimulation is applied during different periods of inspiratory activity and this could impact the swallow motor pattern. In Bv, Non-LAR seems very similar to LAR. In panel E, please add the number of animals.

Thank you, this has been fixed.

We used the same optogenetic protocols in the original paper (Huff et al. 2023) and did not observe any changes to the swallow motor patter in relation to the time PiCo was stimulated. The only phase dependent response seen in both control and CIH is when PiCo Is stimulated during inspiration and a swallow is triggered, inspiration will be inhibited. Therefore, we do not believe variability in swallow motor pattern is dependent on the phase of breathing in which PiCo is stimulated.

Biv LAR has a pause in EMG activity before the swallow begins (red arrow pointing to the pause). While Bv Non-LAR does not have this pause, rather the two behaviors converge (red arrow). In order for something to be considered an LAR the pause must be present which is why we separated these two motor patterns.

Figure 1 - Figure Supplement 1. Why do the Authors call the lines "histograms"?

Thank you, this has been fixed. This is a line graph of swallow frequency in relation to inspiration.

Tables:In tables, data are provided as means and standard deviation. Please, specify this in the Method section.

Thank you, the following is listed in the methods section: “All data are expressed as mean ± standard deviation (SD), unless otherwise noted.”

**Reviewer #3 (Public Review):**
In the present study, the authors investigated the effects of CIH on the swallowing and breathing responses to PICO stimulation. Their conclusion is that glutamatergic-cholinergic neurons from PICO are not only critical for the gating of post-inspiratory and swallow activity, but also play important roles in the generation of swallow motor patterns. There are several aspects that deserve the authors' attention and comments, mainly related to the study´s conclusions.The authors refer to PICO as the generator of post-inspiratory rhythm. However, evidence points to this region as a modulator of post-inspiratory activity rather than a rhythmogenic site (Toor et al., 2019 - 10.1523/JNEUROSCI.0502-19.2019; Oliveira et al., 2021 - 10.1016/j.neuroscience.2021.09.015). For example, sustained activation of PICO for 10 s barely affected the vagus or laryngeal post-inspiratory activity (Huff et al., 2023 - 10.7554/eLife.86103).

Yes, we did refer to PiCo as the postinspiratory rhythm generator as defined as Anderson et al. 2016. We base this statement on the following criteria and experiments: In Anderson et al. 2016, we demonstrate that PiCo can be isolated in vitro, that PiCo neurons are activated in phase with postinspiration, and that they are inhibited during inspiration by preBötC neurons via GABAergic mechanisms and not glycinergic mechanisms. We also demonstrate that optogenetically stimulating cholinergic neurons in the PiCo area resets the inspiratory rhythm both in vivo and in vitro. We also show that PiCo when isolated in transverse slices is autorhythmic and that PiCo, like the preBötC in transverse slices can generate respiratory rhythmic activity in vitro and independent of the preBötC. We also demonstrate that PiCo neurons are an order of magnitude more sensitive to opioids (DAMGO) than the preBötC and that local injections of DAMGO into the PiCo area in vivo abolishes postinspiration, and also abolishes the phase delay of the respiratory rhythm. None of these specific rhythmogenic properties have been studied by the Toor study or the Oliveira et al study. Hence, we do not understand why the reviewer cites these studies as evidence for modulation as opposed to rhythmogenic properties. The fact that PiCo is rhythmogenic should not be considered as an “exclusive property”. Specifically, this does not mean that PiCo is also “modulating” the swallow-breathing coordination as we have demonstrated more specifically in the Huff et al study. In the same sentence we also referred to the PreBӧtzinger complex as the inspiratory rhythm generator as defined by Smith et al 1991, and it seems that the reviewer did not object to this reference. But we would like to point out that the same criteria were used to define the preBötzinger complex as we used for PiCo, except that PiCo neurons are better defined than preBötzinger complex neurons. Dbx1 neurons are often used to characterize the PreBötC, but these neurons form a rostrocaudal and ventrodorsal column which involves also glia cells and transcends the preBötC. Glutamatergic neurons are everywhere, and so are Somatostatin or Neurokinin neurons. Moreover, the 1991 study was only performed in vitro, and did not include a histochemical analysis. We would also like to point out that the present manuscript is investigating the role of PiCo in swallow and laryngeal behaviors, and not specifically postinspiration. Thus, we are not entirely sure how this comment relates to this manuscript.

The optogenetic activation of glutamatergic and cholinergic neurons from PICO evoked submental and laryngeal responses, and CIH changed these motor responses. Therefore, the authors proposed that PICO is directly involved in swallow pattern generation and that CIH disrupts the connection between PICO and SPG (swallow pattern generator). However, the experiments of the present study did not provide evidence about connections between these two regions nor their possible disruption after CIH, or even whether PICO is part of SPG.

We have edited the text to suggest PiCo modulates swallow motor sequence in addition to the coordination of swallow and breathing. We have also added that further experiments will be necessary to further investigate the connections between PiCo and SPG. But, unfortunately, compared to PiCo, the SPG is much less defined. As already stated above, it cannot be expected that a single study can address all possible open questions. Clearly, more work needs to be done outside of this study to answer all of these questions, which makes this an exciting area of research.

CIH affects several brainstem regions which might contribute to generating abnormal motor responses to PICO stimulation. For example, Bautista et al. (1995 - 10.1152/japplphysiol.01356.2011) documented that intermittent hypoxia induces changes in the activity of laryngeal motoneurons by neural plasticity mechanisms involving serotonin.

Yes, we thank the reviewer for this comment and we agree that CIH effects multiple brainstem regions. We stated in the manuscript that we are measuring changes in two muscle complexes which spread among three motor neuron pools: hypoglossal nucleus, trigeminal nucleus, and nucleus ambiguus. We have added a discussion on laryngeal activity in the presence of acute bouts of extreme hypoxia, acute intermittent hypoxia, as well as chronic intermittent hypoxia.

To support the hypothesis that PICO is directly involved in swallow pattern generation the authors should perform the inhibition of Vglut2-ChAT neurons from PICO and then evoke swallow motor responses. If swallow is abolished when the neurons from this region are inhibited, it would indicate that PICO is crucial to generate this behavior.

Thank you. We would like to clarify: “involvement” does not mean “necessary for”. Confusing this difference has caused much confusion and debate in the field. Just as an example: We can argue in great length whether inhibition is necessary for respiratory rhythmogenesis in vivo, but I think there is no question that inhibition is involved in respiratory rhythmogenesis in vivo. But to avoid any confusion, we have changed the text to suggest PiCo is involved in the modulation of swallow motor sequence. We agree various additional inhibition experiments are necessary to explain if PiCo is also a necessary component of the SPG, but this is not the question we have set out to address in this study. To specifically target PiCo we must not only inhibit Vglut2 neurons but neurons that express both ChAT and Vglut2. To our knowledge there are no inhibitory DREADD or opsin techniques for cre/FlpO to specifically target these neurons. As stated above, non-experts in the field do not appreciate this technical nuance. However, we have begun to develop novel techniques necessary to inhibit these specific neurons which will be published in the future.

In almost all the data presented, the authors observed different patterns of changes in the motor submental and laryngeal responses to PICO activation, including that animals submitted to CIH (6%) presented a "normal" motor response. However, the authors did not discuss the possible explanations and functional implications of this variability.

We agree that it would be helpful to understand the functional implications of these alterations in swallow-related motor activation, however since we are not using any tools to measure or evaluate functional activity it would be inappropriate to make suggestions of this type without any data to back up our conclusion. This is why we have not included any functional implications. We have added the following to the manuscript. “While fine wire EMG studies are an excellent evaluation tool to observe temporal motor pattern of sequential swallow related muscles; it must be combined with tools such as videofluoroscopic swallow study (VFSS) and/or high resolution manometry (HRM) in order to characterize the functional significance of these alterations to the swallow motor pattern shown in this study (Park et al., 2017). Since the preparation in this study utilizes only fine wire EMGs we are not able to evaluate or comment on the functional significance of the variable swallow motor patterns.”

In Figure 4, the authors need to present low magnification sections showing the PICO transfected neurons as well as the absence of transfection in the ventral respiratory column. The authors could also check the scale since the cAmb seems very small.

Thank you, added different histology images to have a more comparable cAmb. As well as added lower magnification to show absence of transfection in the VRC.

Finally, the title does not reflect the study. The present study did not demonstrate that PICO is a swallow pattern generator.

We have also changed the title to say: Chronic Intermittent Hypoxia reveals the role of the Postinspiratory Complex in the mediation of normal swallow production